# CAB-KGC: CONTEXT-AWARE BERT FOR KNOWLEDGE GRAPH COMPLETION

## ABSTRACT

Knowledge graph completion (KGC) seeks to predict missing entities (e.g., heads or tails) or relationships in knowledge graphs (KGs), which often contain incomplete data. Traditional embedding-based methods, such as TransE and ComplEx, have improved tail entity prediction but struggle to generalize to unseen entities during testing. Textual-based models mitigate this issue by leveraging additional semantic context; however, their reliance on negative triplet sampling introduces high computational overhead, semantic inconsistencies, and data imbalance. Recent BERT-based approaches, like KG-BERT, show promise but depend heavily on entity descriptions, which are often unavailable in KGs. Critically, existing methods overlook valuable structural information in the KG related to the entities and relationships. To address these challenges, we propose Context-Aware BERT for Knowledge Graph Completion (CAB-KGC), a novel model that utilizes contextual information from linked entities and relations within the graph to predict tail entities. CAB-KGC eliminates the need for entity descriptions and negative triplet sampling, significantly reducing computational complexity while enhancing performance. Additionally, we introduce the Evaluation based on Distance from Average Solution (EDAS) criterion to the KG domain, enabling a more comprehensive evaluation across diverse metrics. Our experiments on standard datasets, including FB15k-237, WN18RR, CoDEx-S, and ConceptNet100K, demonstrate that CAB-KGC outperforms state-of-the-art methods on three datasets. Notably, CAB-KGC achieves improvements in Hit@1 of 6.88%, 14.32%, and 17.13% on WN18RR, CoDEx-S, and ConceptNet100K, respectively. Furthermore, EDAS rankings establish CAB-KGC as the top-performing model, highlighting its effectiveness and robustness for KGC tasks.

## 1    INTRODUCTION

Knowledge graphs (KGs) are structured representations of information about entities $E$ and relationships $R$, expressed as triplets $T$, each triplet consisting of a head, a relationship, and a tail $(h, r, t)$. Google introduced the concept of KG, as a comprehensive framework for storing and organizing interconnected data on entities and relationships (Chen et al., 2020). While there are many publicly available large KGs such as WordNet (Miller, 1995), DBpedia (Auer et al., 2007), FreeBase (Bollacker et al., 2008), and UMLs (Liang et al., 2024) in use, some domain-specific KGs, such as NATION, LOCATION, and SPORTS, are also employed in research and testing. (Liang et al., 2024). Integrated with deep learning and big data, KGs are increasingly being utilized in intelligent systems for tasks like question-answering, recommendation, and natural language processing (Li & Hou, 2017). KGs, despite their effectiveness, are however often incomplete, missing data in the form of entities (heads/tails) or relationships between the entities, which limits their potential as gaps in knowledge reduce the performance of downstream applications (Ding & Jia, 2018). To address this, the Knowledge Graph Completion (KGC) or Link Prediction problem aims at predicting these missing entities $E$ and their interrelationships $R$ thereby enhancing KG density (Lin et al., 2015). In this work, we specifically focus on tail entity prediction, which involves inferring the missing tails in incomplete triplets. Tail prediction is generally more challenging than relation prediction due to the higher diversity and volume of unique entities compared to relationships set in most KGs.

Various strategies have been proposed to address KGC, broadly categorized into knowledge graph embedding (KGE) methods, text-based methods, and large language model-based approaches. KGE-based methods represent entities and relationships as low-dimensional vectors (embeddings) and use them to predict the likelihood of an entity or relationship being in a triplet (Vashishth et al., 2020). However, these methods can only learn embeddings for entities and relations seen in the training data (Xie et al., 2016). When new entities or relationships appear during testing, their embeddings remain randomly initialized, leading to the "unseen entity and relationship" problem. This significantly impacts model performance in tail entity $(h, r, ?)$ prediction. Furthermore, these models learn fixed representations for entities and relations, despite potential variations in meaning or importance across different triples(Yao et al., 2019).

Text-based models like StaR and ConvE utilize neighborhood information from the knowledge graph, utilizing star geometries or convolutions to improve predictions, but they face limitations related to scalability, model complexity, and the ability to handle highly diverse or complex relational structures or may rely on computationally expensive techniques like path ranking and random walks (Bordes et al., 2013; Yang et al., 2014; Vashishth et al., 2020). Modern textual-BERT-based models like KG-BERT and MEM-KGC leverage BERT-based language models, enhancing KGC performance by incorporating semantic knowledge from large text corpora (Choi et al., 2021; Yao et al., 2019; Wang et al., 2022). However, they also struggle in the absence of explicit KG structure, particularly for lexically similar candidates, and often require entity descriptions, which are not available in many KG datasets.

Recently, LLM-based models have adopted tailored prompting techniques to improve prediction accuracy (Zhang et al., 2020b). However, these prompt-based approaches face challenges in efficiently incorporating large-scale KG-specific facts into prompts and struggle when dealing with knowledge graphs from domains that the LLMs have limited prior knowledge of (Wei et al., 2023). A very recent work, DIFT (Liu et al., 2024) finetunes generative LLMs, like LLaMA-2-7B, for KGC by combining it with an embedding-based model. Aside from promping issues, such approaches are of high computational cost associated with training large LLMs like LLaMA-2-7B, which can be resource-intensive and impractical for many real-world applications. These limitations of current models underscore the necessity for a more efficient and trainable KGC approach that eliminates reliance on entity descriptions and negative triplet sampling. Instead, such a model should leverage structural properties of KGs, including contextual information, to ensure broad applicability across diverse knowledge graphs.

Furthermore, current KGC models commonly use assessment metrics such as mean rank (MR), mean reciprocal rank (MRR), and Hit@K for evaluating and ranking models, but they come with notable limitations. In particular, most of the KGC models exhibit inconsistencies across these evaluations (Li et al., 2024). For instance, a model may excel in MRR and Hit@3 while performing poorly in Hit@1. These inconsistencies complicate the identification of a *single best-performing approach across different datasets*. MRR, which emphasizes the rank of the first correct prediction, often disproportionately rewards models for achieving high precision at the top of the ranking, ignoring overall coverage. Conversely, Hits@K, which measures the proportion of correct predictions within the top K ranks, can mask issues related to ranking quality by focusing on whether correct answers exist in a limited range rather than their exact position. Both metrics can be overly influenced by dataset-specific characteristics, such as entity frequency, sparsity, or graph topology, leading to biased or misleading performance assessments. Moreover, these metrics are typically evaluated in isolation, requiring manual interpretation and weighting to understand trade-offs between precision and coverage. Their reliance on single-ground-truth assumptions can further distort results in cases where multiple valid predictions exist but are not explicitly captured in the dataset. These limitations highlight the need for an *aggregated, dataset-agnostic metric*, that balances precision and coverage while reducing biases inherent to individual datasets.

To address the limitation of current KGC approaches, we propose a Context-Aware BERT for Knowledge Graph Completion (CAB-KGC) approach that extracts contextual information from the *entities* and *relationships* associated with the head entity and the relation in question. This context is then integrated with the BERT model to enhance the prediction of tail entities. Additionally, we introduce the use of an alternative evaluation criterion in KG domain, Evaluation based on Distance from Average Solution (EDAS), which allows for a more nuanced analysis and ranking of methodologies. Our results indicate that CAB-KGC outperforms existing KGC methods, including KG-BERT (Yao et al., 2019), MTL-KGC (Kim et al., 2020), and MEM-KGC (Choi et al., 2021), in terms of Mean Reciprocal Rank (MRR) and Hit@K, specifically for k = 1 and 3 (Tapley Hoyt et al., 2022). The EDAS evaluation further corroborates the performance of CAB-KGC, positioning it as a leading approach in the field. To summarize, this study makes the following **contributions** to the KG domain:

- We introduce CAB-KGC approach for tail prediction, which uniquely leverages head entity context and relationship context — distinctive structural KG features overlooked in prior methods — to achieve state-of-the-art performance on multiple datasets.

- CAB-KGC eliminates reliance on entity descriptions unlike other models, enabling applicability to datasets with sparse descriptive information.

- By utilizing cross-entropy loss, CAB-KGC avoids dependence on negative sample training, thereby enhancing training speed and resilience against negative sample selection.

- Our work introduces the use of Evaluation based on Distance from Average Solution (EDAS) methodology to address ranking challenges in KGC arising from variability across performance metrics, providing a reliable comparative framework.

- Extensive experiments on various benchmark datasets demonstrate that CAB-KGC reliably excels in tail entity prediction.

## 2 RELATED WORK

KGC is a significant challenge in Knowledge Graph Analysis (KGA), which explicitly predicts missing data in KG. The missing data may be head $(?, r, t)$, tail $(h, r, ?)$, or relationship $(h, ?, t)$.The existing literature provides multiple KGC methods, which can be classified into two that have gained considerable attention: Knowledge Graph Embedding (KGE) and language model based approaches.

Embedding-based techniques (KGE) capture a graph's underlying semantics by representing entities and relationships as continuous vectors. These can further categorize into translation distance-based methods and semantic or textual matching-based methods (Wang et al., 2017). The first group includes TransE (Bordes et al., 2013), RotatE (Sun et al., 2019), and others approaches (Nickel et al., 2011; Trouillon et al., 2016; Wang et al., 2019; Vashishth et al., 2020; Zhang et al., 2020a), which train a scoring function to measure the distance between entities, modeling interactions as translation operations. In contrast, semantic or textual matching methods, such as DistMult (Yang et al., 2014), rely on scoring functions that compute the similarity between entities and relationships. These methods identify the most likely tail entity during testing by evaluating scores across all entities that share a specific head and relationship.

In translation-based methods, relationships play a crucial role as they facilitate translations between entities $(h, t)$, allowing the scoring function to compute triplet scores. TransE, a foundational model in this category, represents entities and relationships in a shared d-dimensional vector space and calculates triplet scores using the function $TransE_f(h, t) = -||h + r - t||$, where $h, r$, and $t$ are vectors in $R^d$ (Bordes et al., 2013). It assumes that $h + r \cong t$, enabling it to efficiently model simple relational patterns. However, TransE struggles with complex relationship types such as $1 - N$, $N - 1$, and $N - N$ mappings (Wang et al., 2014). RotatE addresses this limitation by representing entities and relationships in a complex vector space $C^d$ and employing rotations to model relationships, with its scoring function defined as $RotatE_f(h, r, t) = -||h \circ r - t||$ (Sun et al., 2019). Building on this, DualE combines rotation and translation operations in dual quaternion space to improve expressiveness (Cao et al., 2021). Similarly, HAKE uses a polar coordinate system to capture semantic hierarchies, offering enhanced capability for modeling hierarchical relations (Zhang et al., 2020a). In contrast, semantic-based approaches like DistMult utilize matrix-based operations to assess the similarity between entities and relationships. The scoring function for DistMult is expressed as $DistMult_f(h, r, t) = h^T diag(r)t$, where $h, r$, and $t$ are elements of $R^d$ (Yang et al., 2014). This method excels at capturing compositional semantics through matrix multiplication. However, despite its effectiveness in modeling some types of relationships, DistMult is inherently limited in its ability to predict missing KG information accurately due to its symmetric nature and inability to handle asymmetric relationships effectively (Zhang et al., 2020a).

Some recent studies leverage text-based models for KGC (Devlin et al., 2019). Among these, KG-BERT, proposed by (Yao et al., 2019), was the first to utilize BERT for the KGC problem. KG-BERT treats knowledge graph triplets as ordered sequences and applies a binary cross-entropy loss function to classify these sequences. While KG-BERT achieves reasonable performance in inferring valid triplets, it falls short in ranking tasks, particularly in metrics such as Hits@1 and Hits@3, when compared to state-of-the-art approaches. Building on this foundation, another study (Kim et al., 2020) introduced a multi-task learning framework (MT-DNN) (Liu et al., 2019), combining KGC with auxiliary tasks such as relationship prediction and an enhanced evaluation mechanism. This approach demonstrated improved overall performance, leveraging multi-task learning to outperform KG-BERT. Another model, MEM-KGC (Choi et al., 2021) employs a masked language model strategy by masking the tail entity in a triplet and predicting it using relationships and available entity descriptions. While MEM-KGC demonstrates notable improvements, its reliance on textual descriptions restricts its applicability across diverse KGs. Most of these text-based KGC models require entity and relationship descriptions to predict missing entities. They however face limitations when applied to datasets like UMLs, National, Location, DBpedia, Family, Kinship, KG20C (Liang et al., 2024) and ConceptNet (Speer et al., 2017), which often lack detailed entity descriptions.

Some hybrid approaches have also been explored. For instance, a hybrid technique (Zhang et al., 2020b) combines LLMs with traditional KGE methods, encoding KG entities and relationships in a shared vector space to serve as initial embeddings. These embeddings are refined through traditional models like TransE or RotatE, enabling improved loss detection and iterative optimization. Similarly, DIFT(Liu et al., 2024) leverages fine-tuned LLaMA-2-7B models, guided by discrimination instructions from embedding-based models. This approach employs a knowledge adaptation module to enhance the LLM's reasoning capabilities within the KG domain. While DIFT achieves state-of-the-art performance, its reliance on computationally intensive LLMs like LLaMA-2-7B presents a significant drawback, making it less practical for resource-constrained applications.

To summarize, existing KGC methodologies face several limitations. Embedding-based models, such as TransE and RotatE, struggle with unseen entities and relationships. Their reliance on pre-defined embeddings restricts adaptability to new information, reducing generalizability across diverse knowledge graph structures. Additionally, these methods often require extensive negative sampling during training, significantly increasing computational costs, particularly on

large datasets. Similarly, text-based and LLM-based approaches, including KG-BERT and MEM-KGC, are heavily dependent on the availability of descriptive entity information, which is often sparse or inconsistent in many datasets. Moreover, approaches like KICGPT and DIFT require complex prompt engineering and fine-tuning, adding to their computational and resource intensity.

A key shortcoming of current KGC approaches is the under-utilization of structural features inherent in knowledge graphs. Most methods fail to incorporate meaningful contextual information from the graph's structure, such as neighboring entities and relationships associated with given head entity and relationship. Table 1 summarizes existing KGC approaches alongside the structural features they utilize, highlighting the gaps in leveraging these critical components. Essential structural features that can provide significant contextual knowledge include the head context-entities (information about neighboring entities), the head context-relations (information about neighboring relations), and the relationship context (details involving the relation ).

The novelty of our proposed CAB-KGC model lies in its comprehensive integration of these three structural components. By capturing the neighboring entity information, surrounding relation details, and relationship context, CAB-KGC provides a richer contextual understanding that enhances prediction accuracy and generalization capabilities. Unlike existing methods, it combines textual and structural information, offering a significant advancement in predictive performance for KGC. The next section presents the design and implementation of the CAB-KGC model.

Table 1: Methods and Components of KG Currently Used in KGC, Considered (●), Not Considered (○). The components are the *Head context-entities*, the *Head context-relations*, and the *Relationship context*. The novelty of the proposed CAB-KGC method lies in its unique integration of the three structural features of KG i.e., the neighbor entity information, the surrounding relation information, and the relationship context, enabling enhanced contextual understanding and superior performance in KGC.

| Methods Exploiting KG Structural Feature | Head context-entities | Head context-relations | Relationship context |
|---|---|---|---|
| TransE (Bordes et al., 2013) | ○ | ○ | ○ |
| DistMult (Yang et al., 2014) | ○ | ○ | ○ |
| RotatE (Sun et al., 2019) | ○ | ○ | ○ |
| HAKE (Zhang et al., 2020a) | ○ | ○ | ○ |
| DualE (Cao et al., 2021) | ○ | ○ | ○ |
| Pre-train-KGC (Lv et al., 2022) | ○ | ○ | ○ |
| TuckER (Wang et al., 2019) | ○ | ○ | ○ |
| TransH (Wang et al., 2014) | ○ | ○ | ○ |
| TransD (Ji et al., 2015) | ○ | ○ | ○ |
| ComplEx (Trouillon et al., 2016) | ○ | ○ | ○ |
| ConvE (Dettmers et al., 2018) | ○ | ○ | ○ |
| KG-BERT (Yao et al., 2019) | ○ | ○ | ○ |
| MEM-KGC (Choi et al., 2021) | ○ | ○ | ○ |
| NNKGC (Wang et al., 2022) | ● | ○ | ○ |
| SimKGC (Wang et al., 2022) | ○ | ○ | ○ |
| Proposed CAB-KGC | ● | ● | ● |

## 3 METHODOLOGY

This section describes the CAB-KGC technique, which predicts tail entities given a head entity and a relationship from a knowledge graph. Before illustrating the model, a brief description of the problem is provided.

**Problem Formulation** (see Table 2 for all relevant notations): Consider a knowledge graph $G(E, R, T)$ as a collection of triplets $T$, where each triplet is given as $(h, r, t)$. Here, $h \in E$ is the head entity, $t \in E$ is the tail entity, and $r \in R$ represents the relationship between them. Given an incomplete triple $(h, r, ?)$, CAB-KGC model predicts the missing tail $t$ (represented by ? in the incomplete triple) .

To achieve accurate tail prediction, the proposed CAB-KGC method extracts contextual information from the knowledge graph surrounding the head entity and the relationship in question, and leverages this context within a BERT model to learn and generate precise predictions. Figure 1 provides a detail overview of the CAB-KGC model. Given a head $h$ and a relationship $r$, our model predicts the tail entity $t$ , in the following steps:

**Step 1. Extract Head Context** $H_c$: To extract the contextual information for the head, i.e. $H_c$, we first identify the relationships $r$ that are associated with the head entity $h$, i.e., the relationship neighborhood $\mathcal{R}(h)$. If $k$ relationships are associated with the head $h$ from the set $R$ of all relationships $r_i$ in the graph $G$, then:

$$\mathcal{R}(h) = A_{i=1}^{k} \left( \{r_i \mid (h, r_i, e_j) \in T, e_j \in E\} \right) \tag{1}$$

where $A(\cdot)$ is some aggregator function (in our case, it is the concatenation operation $\|$ ), $T$ is the set of training triplets, $E$ is the set of all entities and $r_i$ represents each relation associated with $h$.

Table 2: Mathematical Notations and Symbols

| Notation | Description | Notation | Description |
|---|---|---|---|
| $e, e_i$ | entity or node | $r, r_j$ | relationship |
| $h$ | head entity node | $t$ | tail entity node |
| $E$ | Entities Set | $R$ | Relationships Set |
| $H_c$ | Head ($h$) context | $R_c$ | Relationship context |
| $T$ | Set of Triplets | $N_T$ | Total number of triplets |
| $\mathcal{R}(h)$ | Relations connected to head $h$ | $\mathcal{E}(h)$ | Entities connected to head $h$ |
| $P(t_i \mid h_i, r_i)$ | Tail $t_i$ probability given head $h_i$ and relationship $r_i$ | $\text{rank}_i$ | Rank of the true tail entity $t_i$ in the prediction |
| $\downarrow$ | Results: Lower is better | $\uparrow$ | Results: Higher is better |
| $Xi$ | Individual result of each component | WPav | Weighted Positive Average |
| N(WPav) | Normalized Weighted Positive Average | N(WNav) | Normalized Weighted Negative Average |
| $M$ | The final aggregated evaluation score | $A(\cdot)$ | Any Aggregator Function |
| WNav | Weighted Negative Average | | |

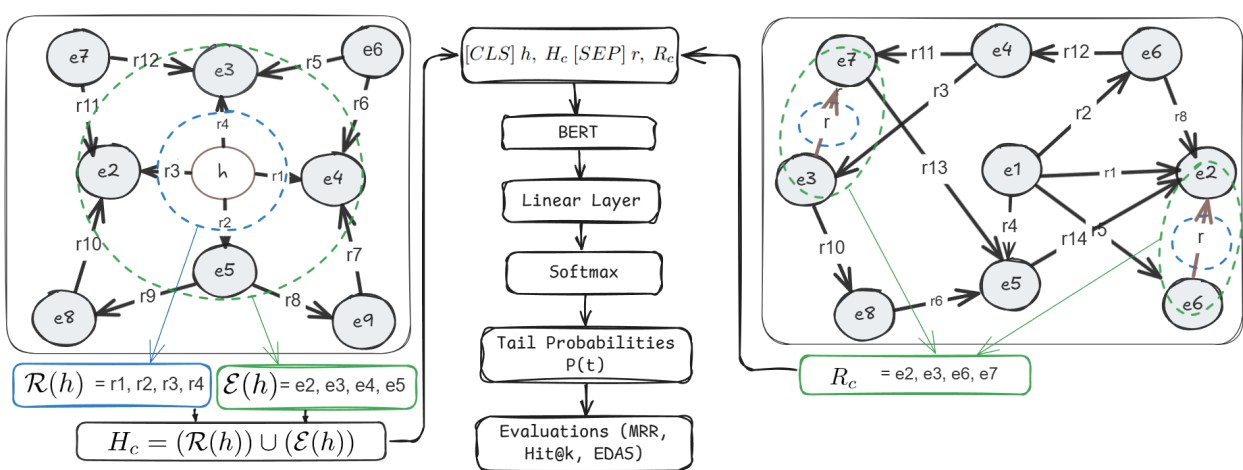

Figure 1: A concise overview of the CAB-KGC model pipeline for predicting the tail entity, given a head entity $h$ and a relationship $r$. The box on the left illustrates the calculation of head context $H_c$. $H_c$ is formed as a union of $\mathcal{R}(h)$ and $\mathcal{E}(h)$. Here, $\mathcal{R}(h)$ is the set of all relations ($r1$, $r2$, $r3$ and $r4$) involving the head entity $h$, while $\mathcal{E}(h)$ is the set of all neighboring entities ($e2$, $e3$, $e4$, and $e5$) directly related to $h$. The box on the right shows the calculation of relationship context $R_c$. $R_c$ comprises the set of all entities ($e3$, $e7$, $e2$, and $e6$) associated via relationship $r$. These contextual features — $H_c$ and $R_c$ — alongside $h$ and $r$ are then fed as input to the BERT model as depicted in the middle of the figure. The BERT model, combined with a linear classifier and softmax, generates probabilities for tail entities.

Next, we find the entities $e$ that are neighbors (i.e., have a direct connection) with the head entity $h$, i.e., entity neighborhood $\mathcal{E}(h)$ using the identified relationships in $\mathcal{R}(h)$. Assuming $m$ neighbor entities, $\mathcal{E}(h)$ is expressed as:

$$\mathcal{E}(h) = A_{i=1}^m \left( \{e_i \mid (h, r_j, e_i) \in T, \ r_j \in R\} \right) \tag{2}$$

where $R$ is the set of all relations and $e_i$ represents each entity directly associated with $h$.

The head context $H_c$ is then calculated as the union of the connected relations $\mathcal{R}(h)$ and the neighbor entities $\mathcal{E}(h)$, as shown below in Equation 3.

$$H_c = (\mathcal{R}(h)) \cup (\mathcal{E}(h)) \tag{3}$$

Conceptually, the head context $H_c$ represents the entity-centric local knowledge. It pulls out:

1. Entities directly linked to the head entity $\mathcal{E}(h)$: This captures local neighborhood information, essentially providing a "snapshot" of the immediate surroundings of the head entity in the graph. This structural information helps the model identify how the head entity $h$ is positioned within its local neighborhood (e.g., is it a hub, part of a chain, etc.).

2. Relations associated with these neighboring entities $\mathcal{R}(h)$: By including the relationships between the head entity and its neighbors, we bring in a semantic understanding of the types of relationships the entity $h$ is generally involved in.

Therefore, the head context's role is to provide **specificity**. By narrowing the focus to the immediate surroundings of the head entity $h$, it helps the model differentiate between plausible tail entities based on the specific neighborhood structure.

**Step 2. Extract Relationship Context** $R_c$: To acquire the relationship context $R_c$, we identify all the entities associated with the operational relationship $r$ in the knowledge Graph $G$. $R_c$ is given as:

$$R_c = A_{i,j=1}^l \left(\{e_i, e_j \mid (e_i, r, e_j) \in T\}\right) \tag{4}$$

where $e_i, e_j$ are entities in $E$ connected by the relation $r$.

Conceptually, the relation context $R_c$ represents relation-centric global knowledge. It pulls out all entities connected via the given relation $r$. Instead of focusing on the head entity's neighborhood, this provides a global perspective on the nature of the relation $r$ itself and the patterns or clusters involving the relation $r$. This helps the model learn global patterns about how this relation typically connects entities across the entire KG (e.g., relation *capital_of* connects countries and cities). Relation context $R_c$ also provides the model with the knowledge of the distribution or variability of the manifestation of the relation $r$ in the KG. This helps the model refine predictions by learning how the relation $r$ operates beyond the local context of the head entity.

In other words, the relation context's role is to provide **generalization**. By considering global patterns of the relation, it acts as a regularizer, ensuring that the model aligns with broader relational constraints in the KG.

Together, the head and relation contexts allow the model to leverage local specificity (head context) and ensure global consistency (relation context). This innovation solves the "description sparsity" problem in dealing with datasets without descriptions.

**Step 3. Prepare Input Sequence for BERT Classifier**: The contextual information extracted in above steps forms the input to BERT. Specifically, the input sequence contains h, $H_c$ from Equation 3, $r$, and $R_c$ from Equation 4, as shown below:

$$\text{Input Sequence} = [CLS]\ h,\ H_c\ [SEP]\ r,\ R_c \tag{5}$$

where [CLS] is BERT's classifier token and [SEP] is the separator token.

**Step 4. Predict and train with BERT Classifier**: A classification layer is added on top of the BERT model, which aims to classify all the entities from Entity set $E$. Once the BERT classifier receives the input, it processes it through various transformer layers, provides a contextualized representation of each token and uses that to classify the input. The classifier model predicts the tail entity by employing a softmax function over the output pooled representation to calculate the probability for all the available tail entities. The input-output description of the model is given as:

$$(\text{BERT Output}) = \text{BERT(Input Sequence)} \tag{6}$$
$$P(t \mid h, r) = \text{softmax}(W \cdot \text{BERT Output}) \tag{7}$$

where $W$ is a learnable weight matrix.

Putting above equations together, the CAB-KGC model can be expressed as:

$$\text{CAB-KGC}(t \mid h, r) = \text{softmax}(W \cdot \text{BERT}(h, H_c, r, R_c)) \tag{8}$$

The CAB-KGC model is trained using cross-entropy loss, which compares the probability distribution of the predicted label with the true label for the tail entity. The cross-entropy loss is given by:

$$L = -\sum_{i=1}^N y_i \log P(t_i \mid h, r)g \tag{9}$$

In this equation the one-hot encoded true label for the tail object $t_i$ is indicated as $y_i$. The predicted probability for the true tail entity could be denoted as $P(t_i \mid h, r)$, where $h$ is the head and $r$ is the relation.

## 3.1 EXPERIMENTS

**Datasets:** We assess the proposed CAB-KGC model on various commonly used KG datasets. The survey work by Liang et al. (2024) provides a comprehensive collection of freely available datasets for KGC and reasoning tasks via their GitHub repository [1]. These datasets are briefly explained here:

- FB15k-237 (Toutanova & Chen, 2015) is an updated version subset of the FB15k dataset, where the inverse triplets have been removed to increase the difficulty of the KGC. It has 14,541 entities, 237 relationships, 272,115 training triplets, 17,535 validation triplets, and 20,466 testing triplets.
- WN18RR, is the subset of WN18 where the reverse triplets are removed, making it more complex for the models to incorporate the problem of KGC (Miller, 1995). It contains 40,943 entities, 11 relationships, 86,835 training triplets, 2,924 validation triplets, and 2,824 testing triplets.
- The CoDEx-S KG dataset, sourced from Wikidata, is an inclusive benchmark covering multi-domain, diverse and interpretable content, offering a thorough and authentic testing environment (Safavi & Koutra, 2020). It has 2,034 entities, 42 relationships, 32,888 training triplets, 3,654 validation triplets, and 3,656 testing triplets.
- ConceptNet100K, is a sparse subset of ConceptNet KG, which is a graph version of the Open Mind Common Sense project (Speer et al., 2017). It consists of semantically rich, interconnected nodes that encode human-like reasoning about everyday concepts and relationships. It includes 78,334 entities, 34 relationships, 100,000 training triplets, 1,200 validation triplets, and 1,200 testing triplets.

**Hyperparameters:** The experiments used a batch size of 16 and a learning rate of 5e-5, Adam as the optimizer and cross-entropy as the loss function. The experiments were accomplished on an NVIDIA GeForce RTX 3090 GPU with 24 GB of memory. Training for the CAB-KGC model and its variants was halted once evaluation metrics stabilized to the third decimal place, typically within 30 epochs.

**Evaluation:** Various standard evaluation metrics in KGC, as given in Equation 10, such as MRR, and Hit@k, are utilized to assess the performance of the proposed method and other state-of-the-art approaches

$$\text{MRR} = \frac{1}{N}\sum_{i=1}^{N}\frac{1}{\text{rank}_i} \quad ; \quad \text{Hits@k} = \frac{1}{N}\sum_{i=1}^{N}\mathbf{1}(\text{rank}_i \leq k) \tag{10}$$

where $\text{rank}_i$ is the correct entity rank position in the descending order sorted list of predicted scores for the $i$-th triplet. The function $\mathbf{1}(\text{rank}_i \leq k)$ is a ranking function that outputs one if the true entity is ranked within the top $k$ predictions and 0 otherwise.

**Evaluation based on Distance From Average Solution (EDAS) :** The alternative evaluation based on distance from average solution (EDAS) assessment criteria we use in our study is a flexible option that assesses methods based on a wider range of parameters and provides a more comprehensive performance assessment (Ghorabaee et al., 2015). Using multiple ranking criteria, EDAS overcomes the constraints of MRR and Hit@K, ensuring a complete evaluation of models across all relevant ranks. EDAS provides a final rank for each method, making it simple for the reader to evaluate and rank it (Torkayesh et al., 2023). It assesses the schema by taking into account the average solution it generates. The average solution is determined by calculating the sum of positive and negative distances from the mean values. Results with the most outstanding aggregated validation score $M$ value correspond to the highest-ranked results. The average value for each criterion, whether beneficial (MRR and Hit@k higher is better) or non-beneficial (MR lower is better), is computed using the following formula:

$$\text{Average Value} = \frac{\sum_{i=1}^{n} X_i}{n} \tag{11}$$

In the given equation, Xi denotes the value of each evaluation criterion (MR, MRR, or Hit@k) for $i$-th method, while n quantifies the total number of values. Next, for each method, the positive distance from average (PDA) and negative distance from average (NDA) are calculated as follows:

Non-Beneficial Criteria:

$$\begin{cases} \text{PDA}_{ij} = \dfrac{\max(0, \text{Avg}_j - X_{ij})}{\text{Avg}_j} \\ \text{NDA}_{ij} = \dfrac{\max(0, X_{ij} - \text{Avg}_j)}{\text{Avg}_j} \end{cases}$$

Beneficial Criteria:

$$\begin{cases} \text{PDA}_{ij} = \dfrac{\max(0, X_{ij} - \text{Avg}_j)}{\text{Avg}_j} \\ \text{NDA}_{ij} = \dfrac{\max(0, \text{Avg}_j - X_{ij})}{\text{Avg}_j} \end{cases} \tag{12}$$

---

[1] https://github.com/LIANGKE23/Awesome-Knowledge-Graph-Reasoning

Here $X_{ij}$ represents the value of the $j$-th criterion for the $i$-th method. Positive Distance Analysis (PDA) is the measure of the positive deviation from the average for the $i$-th method and $j$-th criterion. Alternatively, negative distance analysis (NDA) is the measure of the negative deviation from the average for the $i$-th method and $j$-th criterion. Additionally, for each approach, the weighted positive and negative distances are computed in the following ways:

$$\text{WPav}_i = \frac{1}{m} \sum_{j=1}^{m} \text{PDA}_{ij}, \quad \text{WNav}_i = \frac{1}{m} \sum_{j=1}^{m} \text{NDA}_{ij} \tag{13}$$

where $m$ represents the total number of criteria. Weighted Positive Average $\text{WPav}_i$ represents the weighted sum of positive distances from the average for the $i$-th method. The weighted sum of negative distances from the average for the $i$-th method is denoted as Weighted Negative Average $\text{WNav}_i$. In the subsequent stage, normalization is performed. For the normalization of the weighted sums, the following equations are employed:

$$N(\text{WPav}_i) = \frac{\text{WPav}_i}{\max(\text{WPav})}, \quad N(\text{WNav}_i) = \frac{\text{WNav}_i}{\max(\text{WNav})} \tag{14}$$

where $N(\text{WPav}_i)$ and $N(\text{WNav}_i)$ denote the normalized weighted positive and negative distances for the $i$-th method respectively. Finally, the evaluation score $M$ for each approach is computed in the following manner:

$$M_i = \frac{1}{2} \left( N(\text{WPav}_i) + (1 - N(\text{WNav}_i)) \right) \tag{15}$$

where $M_i$ denotes the aggregated validation score for the $i$-th method. The values of $M_i$ determine a method ranking. The outcome with the highest $M$ score is given the top rank. A detailed example of EDAS calculations is provided in Appendix A.

**Advantages of EDAS over Conventional Assessment Methods:** The conventional assessment procedures for KGC tasks, such as Mean Reciprocal Rank (MRR) and Hit@k, are effective for computing specific aspects of model performance but have limitations. Using the EDAS metric (Torkayesh et al., 2023) derived from MRR and Hits@K offers below advantages for evaluating and ranking KGC models:

- *Aggregated and Reliable Assessment Metrics:* EDAS combines the strengths of MRR (focusing on ranking precision) and Hits@K (highlighting prediction coverage) into a unified metric. MRR emphasizes the rank of the first correct prediction, while Hits@K accounts for the presence of correct predictions within a top-K range. EDAS ensures neither is overlooked. By leveraging multiple metrics, EDAS reduces the risk of over-fitting to one specific evaluation criterion, offering a more robust measure of a model's ranking ability.

- *Single-Valued Comparative Scoring for Benchmarking:* EDAS provides a consolidated score, making it easier to compare models without needing to interpret multiple metrics (e.g., MRR, Hits@1, Hits@3, Hits@10) individually. This is particularly useful for benchmarking across papers, datasets, or use cases. When interpreting MRR and Hit s@K together, there is often ambiguity about how much weight to assign to each metric. EDAS inherently balances these contributions.

- *Dataset-Agnostic Evaluations:* EDAS enables the evaluation of model performance across diverse datasets with varying characteristics (e.g., size, density, and graph structure). This uniform metric avoids biases introduced by dataset-specific peculiarities, ensuring that results are comparable. Since MRR and Hits@K can be skewed by dataset construction (e.g., sparsity or entity frequency), aggregating them into a single EDAS score smoothens these effects, providing a holistic measure. This may also help control the false optimism in metrics. MRR and Hits@K can sometimes produce overly optimistic results in isolation due to dataset idiosyncrasies or trivial solutions (e.g., predicting frequent entities). EDAS penalizes such biases by balancing performance measures across datasets. Furthermore, by aggregating results across multiple datasets, EDAS ensures that a model's strengths are not over-represented due to favorable dataset characteristics, leading to fairer comparisons.

## 3.2 RESULTS

The experimental findings in Tables 3 and 4 present a comprehensive comparison of CAB-KGC against state-of-the-art methodologies across FB15k-237, WN18RR, CoDEx-S, and ConceptNet100k datasets. Evaluated metrics include Mean Reciprocal Rank (MRR), Hits@1, Hits@3, and Hits@10. Table 5 further provides EDAS rankings, synthesizing performance across these datasets. Below, we summarize key insights and patterns.

**Performance on FB15k-237 and WN18RR** CAB-KGC demonstrates strong performance on WN18RR, achieving an MRR of 0.685, outperforming the prior best model, NNKGC (MRR = 0.674). It also secures the highest Hits@1 (0.637) and competitive Hits@3 and Hits@10 scores (0.687 and 0.737), highlighting its strength in capturing complex

patterns. On FB15k-237, CAB-KGC delivers competitive results but falls slightly behind methods like DIFT and NBFNet, achieving an MRR of 0.350 and Hits@10 of 0.462. This performance gap suggests room for improvement in handling dense, interconnected graph structures.

**Performance on CoDEx-S and ConceptNet100k** CAB-KGC achieves top performance on CoDEx-S and Concept-Net100k, consistently surpassing baselines. On CoDEx-S, it achieves an MRR of 0.55 and Hits@10 of 0.764, outperforming the next-best approach, ComplEx (MRR = 0.46, Hits@10 = 0.646). Similarly, for ConceptNet100k, CAB-KGC achieves an MRR of 0.35 and Hits@10 of 0.546, exceeding BiQUE (MRR = 0.32, Hits@10 = 0.533). These results underscore CAB-KGC's superior ability to integrate contextual and structural information in semantically rich and diverse datasets.

Table 3: Comparison of the proposed and baseline methods on the datasets FB15k-237 and WN18RR. The optimal outcome for each metric is highlighted in bold, while the second-best result is underlined. The circle symbol ○ denotes that the results have been extracted from the study conducted by Wei et al. (Wei et al., 2023), while the symbol □ indicates that the results have been extracted from the study conducted by Yao et al. in (Yao et al., 2019).

| Dataset | FB15k-237 | | | | WN18RR | | | |
|---|---|---|---|---|---|---|---|---|
| Methods | MRR ↑ | Hits@1 ↑ | Hits@3 ↑ | Hits@10 | MRR ↑ | Hits@1 ↑ | Hits@3 ↑ | Hits@10 |
| *Embedding-Based Methods* | | | | | | | | |
| RESCAL (Nickel et al., 2011) ○ | 0.356 | 0.266 | 0.390 | 0.535 | 0.467 | 0.439 | 0.478 | 0.516 |
| TransE (Bordes et al., 2013) ○ | 0.279 | 0.198 | 0.376 | 0.441 | 0.243 | 0.043 | 0.441 | 0.532 |
| TuckER (Wang et al., 2019) ○ | 0.358 | 0.266 | 0.394 | 0.544 | 0.470 | 0.443 | 0.482 | 0.526 |
| ComplEx (Trouillon et al., 2016) ○ | 0.247 | 0.158 | 0.275 | 0.428 | 0.440 | 0.410 | 0.460 | 0.510 |
| ConvE (Dettmers et al., 2018) | 0.631 | 0.239 | 0.350 | 0.491 | 0.460 | 0.390 | 0.430 | 0.480 |
| DistMult (Yang et al., 2014) ○ | 0.241 | 0.155 | 0.263 | 0.419 | 0.430 | 0.390 | 0.440 | 0.490 |
| RotatE (Sun et al., 2019) ○ | 0.338 | 0.241 | 0.375 | 0.533 | 0.476 | 0.428 | 0.492 | 0.571 |
| CompGCN (Vashishth et al., 2020) ○ | 0.355 | 0.264 | 0.390 | 0.535 | 0.479 | 0.443 | 0.494 | 0.546 |
| HittER (Chen et al., 2021) ○ | 0.344 | 0.246 | 0.380 | 0.535 | 0.496 | 0.449 | 0.514 | 0.586 |
| HAKE (Zhang et al., 2020a) ○ | 0.346 | 0.250 | 0.381 | 0.542 | 0.497 | 0.452 | 0.516 | 0.582 |
| BiQUE (Guo & Kok, 2021) | 0.365 | 0.270 | 0.401 | 0.555 | 0.504 | 0.459 | 0.519 | 0.588 |
| *Text-and Description-Based Methods* | | | | | | | | |
| Pretrain-KGE (Zhang et al., 2020b) ○ | 0.332 | - | - | - | 0.235 | - | - | - |
| StAR (Wang et al., 2021) ○ | 0.263 | 0.171 | 0.287 | 0.452 | 0.364 | 0.222 | 0.436 | 0.647 |
| MEM-KGC (w/o EP) (Choi et al., 2021) ○ | 0.339 | 0.249 | 0.372 | 0.522 | 0.533 | 0.473 | 0.570 | 0.636 |
| MEM-KGC (w/ EP) (Choi et al., 2021) ○ | 0.346 | 0.253 | 0.381 | 0.531 | 0.557 | 0.475 | 0.604 | 0.704 |
| SimKGC(Wang et al., 2022) ○ | 0.333 | 0.246 | 0.363 | 0.510 | 0.671 | 0.585 | **0.731** | **0.817** |
| *LLM-Based Methods* | | | | | | | | |
| ChatGPTzero-shot (Zhu et al., 2024) □ | - | 0.237 | - | - | - | 0.190 | - | - |
| ChatGPTone-shot (Zhu et al., 2024) □ | - | 0.267 | - | - | - | 0.212 | - | - |
| DIFT (Liu et al., 2024) | **0.439** | **0.364** | **0.468** | 0.586 | 0.617 | 0.569 | 0.638 | 0.708 |
| KICGPT (Wei et al., 2023) □ | 0.412 | 0.327 | 0.448 | 0.554 | 0.549 | 0.474 | 0.585 | 0.641 |
| *GNN-Based Method* | | | | | | | | |
| NBFNet (Zhu et al., 2021) | 0.415 | 0.321 | 0.454 | **0.599** | 0.551 | 0.497 | 0.573 | 0.666 |
| NNKGC(Li & Yang, 2023) ○ | 0.338 | 0.252 | 0.365 | 0.515 | 0.674 | 0.596 | 0.722 | 0.812 |
| *Proposed* | | | | | | | | |
| CAB-KGC  (H$_c$ Only) | 0.310 | 0.263 | 0.331 | 0.403 | 0.420 | 0.492 | 0.556 | 0.616 |
| CAB-KGC  (R$_c$ Only) | 0.280 | 0.187 | 0.255 | 0.271 | 0.321 | 0.345 | 0.371 | 0.398 |
| **CAB-KGC** | 0.350 | 0.322 | 0.399 | 0.462 | **0.685** | **0.637** | 0.687 | 0.737 |

Table 4: Comparison of the proposed and baseline methods on the datasets CoDEx-S and ConceptNet100k. The optimal outcome for each metric is highlighted in bold, while the second-best result is underlined. CoDEx-S results are taken from (Safavi & Koutra, 2020) and ConceptNet results are reported in (Guo & Kok, 2021).

| Dataset | CoDEx-S | | | | ConceptNet100k | | | |
|---|---|---|---|---|---|---|---|---|
| Methods | MRR ↑ | Hits@1 ↑ | Hits@3 ↑ | Hits@10 | MRR ↑ | Hits@1 ↑ | Hits@3 ↑ | Hits@10 |
| RESCAL (Nickel et al., 2011) | 0.40 | 0.293 | - | 0.623 | - | - | - | - |
| TransE (Bordes et al., 2013) | 0.44 | 0.339 | - | 0.638 | - | - | - | - |
| TuckER (Wang et al., 2019) | 0.44 | 0.339 | - | 0.638 | - | - | - | - |
| ComplEx (Trouillon et al., 2016) | 0.46 | 0.370 | - | 0.646 | 0.114 | 0.074 | 0.125 | 0.190 |
| ConvE (Dettmers et al., 2018) | 0.44 | 0.343 | - | 0.635 | 0.209 | 0.140 | 0.229 | 0.340 |
| BiQUE (Guo & Kok, 2021) | - | - | - | - | 0.320 | 0.216 | 0.359 | **0.553** |
| **CAB-KGC** | **0.55** | **0.423** | **0.647** | **0.764** | **0.35** | **0.253** | **0.400** | 0.546 |

**Comparative Ranking Analysis (EDAS)** The EDAS framework evaluates KGC methods holistically by normalizing positive (WPav) and negative (WNav) distances across metrics. Detailed calculations of these values are available in Appendix A, Table 7. We exclude missing values for any criterion metrics associated with a method to ensure consistency and fairness in the computations. CAB-KGC ranks highest (Rank 1), demonstrating robust performance across datasets. DIFT (Rank 2) and NNKGC (Rank 3) follow closely, while conventional embedding models like RESCAL (Rank 14) and DistMult (Rank 18) show weaker results. Text-based methods, such as MEM-KGC (Rank 8), outperform basic embedding models, but CAB-KGC's balanced use of textual and structural features provides a clear advantage. **Insights and Hypothesis Validation** The findings strongly indicate that CAB-KGC outperforms traditional KGE, textual techniques, and LLM procedures across various datasets. The main hypothesis behind CAB-KGC's design to integrate relational and neighboring contextual information with BERT, significantly enhances tail entity predictions. This is evident from its consistent improvements in MRR, Hits@k, and EDAS metrics across all datasets. While FB15k-237 performance suggests opportunities for optimization in dense networks, CAB-KGC demonstrates exceptional flexibility and accuracy, establishing itself as a leading approach for knowledge graph completion.

## 4 ABLATION

The ablation study (see Table 3 )assesses the contributions of the Head Context ($H_c$) and Relation Context ($R_c$) components in the CAB-KGC model on FB15k-237 and WN18RR datasets. The $H_c$-Only configuration, encapsulating the adjacent entities and relationships only, attains reasonable performance with an MRR of 0.310 and Hits@10 of 0.403 on FB15k-237, and an MRR of 0.420 and Hits@10 of 0.616 on WN18RR. The $R_c$-Only, which utilizes global relational patterns, shows inferior performance compared to $H_c$-Only, achieving an MRR of 0.280 and Hits@10 of 0.271 on FB15k-237, and an MRR of 0.321 and Hits@10 of 0.398 on WN18RR, highlighting the constraints of depending exclusively on global context.

## 5 CONCLUSION

The CAB-KGC method presents an accurate and efficient method for KGC by employing neighboring contextual information of the head entity and the relation in question. It overcomes the constraints of current KGE-based methods and LLM mechanisms, illustrating improvements across diverse datasets using multiple metrics. The results demonstrated that the proposed approach, CAB-KGC, performed better than the most advanced methods of KGC on multiple benchmark datasets. The performance of our method exceeded that of existing methods in terms of both MRR and Hit@k measures. In particular, in Hit@1 over WN18RR, CoDEx-S, and ConceptNet100K, CAB-KGC has improved over SOTA by 6.88%, 14.32%, and 17.13% respectively. Additionally, CAB-KGC ranks as the top performer across datasets in EDAS ranking. Our ablation studies revealed that including the context of the head and relationship has a significant impact on model performance. This validates the hypothesis regarding the significance of including contextual information about the KG entity and relationship. The results highlight the broader implications of utilizing contextual information in KGC. CAB-KGC improves accuracy and offers a practical and scalable solution by eliminating the de-

Table 5: EDAS rankings of the models.

| Methods | EDAS Rank |
|---|---|
| RESCAL (Nickel et al., 2011) | 14 |
| TransE (Bordes et al., 2013) | 16 |
| TuckER (Wang et al., 2019) | 11 |
| ComplEx (Trouillon et al., 2016) | 17 |
| ConvE (Vashishth et al., 2020) | 10 |
| DistMult (Yang et al., 2014) | 18 |
| RotatE (Sun et al., 2019) | 15 |
| CompGCN (Vashishth et al., 2020) | 12 |
| HittER (Chen et al., 2021) | 20 |
| HAKE (Zhang et al., 2020a) | 13 |
| BiQUE (Liu et al., 2024) | 7 |
| StAR (Wang et al., 2021) | 19 |
| MEM-KGC (w/o EP) (Choi et al., 2021) | 9 |
| MEM-KGC (w/ EP) (Choi et al., 2021) | 8 |
| SimKGC (Wang et al., 2022) | 4 |
| DIFT (Liu et al., 2024) | 2 |
| KICGPT (Wei et al., 2023) | 6 |
| NBFNet (Zhu et al., 2021) | 5 |
| NNKGC (Li & Yang, 2023) | 3 |
| CAB-KGC (Liu et al., 2024) | **1** |

mand for entity descriptions and negative triplet sampling. In future studies, we may utilize more contextual data, such as spatial and temporal information, to potentially expand the techniques' effectiveness. Furthermore, optimizing the model for large-scale KGs can verify its applicability across domains and datasets.The comprehensive CAB-KGC model, integrating $H_c$-Only and $R_c$-Only, attains optimal performance, underscoring these elements' synergistic relationship. This verifies that the integration of $H_c$-only and $R_c$-Only allows the model to utilize both local specificity and global generalization, resulting in enhanced performance in tail entity prediction.

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

## A   EVALUATION BASED ON DISTANCE FROM AVERAGE SOLUTION (EDAS)

This section provides a detailed step-by-step procedure for the EDAS assessment method followed for KGC. The EDAS approach offers significant benefits and guidance to decision-makers when dealing with complex data that includes many variations. In EDAS, the average solution is found by calculating the distance between positive and negative values from the mean. Generally, the final M score with the highest value is the highest-ranked. Supposing this is our final results matrix:

| Position $X_{ij}$ | MR (non-beneficial) | MRR (beneficial) | Hit@1 (beneficial) | Hit@3 (beneficial) | Hit@5 (beneficial) | Hit@10 (beneficial) |
|---|---|---|---|---|---|---|
| Method 1 | $X_{12}$ | $X_{13}$ | $X_{14}$ | $X_{15}$ | $X_{16}$ | $X_{17}$ |
| Method 2 | $X_{22}$ | $X_{23}$ | $X_{24}$ | $X_{25}$ | $X_{26}$ | $X_{27}$ |
| Method 3 | $X_{32}$ | $X_{33}$ | $X_{34}$ | $X_{35}$ | $X_{36}$ | $X_{37}$ |
| Method 4 | $X_{42}$ | $X_{43}$ | $X_{44}$ | $X_{45}$ | $X_{46}$ | $X_{47}$ |
| Method 5 | $X_{52}$ | $X_{53}$ | $X_{54}$ | $X_{55}$ | $X_{56}$ | $X_{57}$ |
| Method 6 | $X_{62}$ | $X_{63}$ | $X_{64}$ | $X_{65}$ | $X_{66}$ | $X_{67}$ |

Table 6: Results matrix with position indicators $X_{ij}$.

**Step 1: Mean values**

$$\text{MR average value} = \frac{X_{12} + X_{22} + X_{32} + X_{42} + X_{52} + X_{62}}{n}$$

$$...$$

$$...$$

$$\text{Hit@10 average value} = \frac{X_{17} + X_{27} + X_{37} + X_{47} + X_{57} + X_{67}}{n}$$

**Step 2: PDA and NDA Calculation** Beneficial PDA and NDA measurements MRR, Hit@1, Hit@3, Hit@5, Hit@10 (higher better):

$$\text{PDA}_{ij} = \frac{\max(0, X_{ij} - \text{Avg}_j)}{\text{Avg}_j}, \quad \text{NDA}_{ij} = \frac{\max(0, \text{Avg}_j - X_{ij})}{\text{Avg}_j}$$

Method 1 MRR (Step 2, Beneficial Example ):

$$\text{PDA (MRR)}_{13} = \frac{\max(0, X_{13} - \text{MRR average value})}{\text{MRR average value}}, \quad \text{NDA (MRR)}_{13} = \frac{\max(0, \text{MRR average value} - X_{13})}{\text{MRR average value}}$$

Non-beneficial MR (lower better):

$$\text{PDA}_{ij} = \frac{\max(0, \text{Avg}_j - X_{ij})}{\text{Avg}_j}, \quad \text{NDA}_{ij} = \frac{\max(0, X_{ij} - \text{Avg}_j)}{\text{Avg}_j}$$

Method 1 MR (Step 2, Non-beneficial Example):

$$\text{PDA (MR)}_{12} = \frac{\max(0, \text{MR average value} - X_{12})}{\text{MR average value}}, \quad \text{NDA (MR)}_{12} = \frac{\max(0, X_{12} - \text{MR average value})}{\text{MR average value}}$$

**Step 3: Weighted PDA and NDA**

$$\text{WPav}_i = \frac{1}{m} \sum_{j=1}^{m} \text{PDA}_{ij}$$

Method 1 (Step 3, Weighted PDA Example):

$$\text{WPav}_1 = \frac{\text{PDA(MR)}_{12} + \text{PDA(MRR)}_{13} + \text{PDA(Hit@1)}_{14} + \text{PDA(Hit@3)}_{15} + \text{PDA(Hit@5)}_{16} + \text{PDA(Hit@10)}_{17}}{n}$$

$$\text{WNav}_i = \frac{1}{m} \sum_{j=1}^{m} \text{NDA}_{ij}$$

Method 1 (Step 3, Weighted NDA Example):

$$\text{WNav}_1 = \frac{\text{NDA(MR)}_{12} + \text{NDA(MRR)}_{13} + \text{NDA(Hit@1)}_{14} + \text{NDA(Hit@3)}_{15} + \text{NDA(Hit@5)}_{16} + \text{NDA(Hit@10)}_{17}}{n}$$

**Step 4: Normalization of Weighted PDA and NDA** In this step, WPav and WNav distances are normalized:

$$N(\text{WPav}_i) = \frac{\text{WPav}_i}{\max(\text{WPav})}, \quad N(\text{WNav}_i) = \frac{\text{WNav}_i}{\max(\text{WNav})}$$

**Step 5: Final M Score Calculation** Compute the final M score and determine the ranking based on the M values. The higher M reflects a superior rank:

$$M_i = \frac{1}{2}\left(N(\text{WPav}_i) + (1 - N(\text{WNav}_i))\right)$$

Table 7 provides a detailed EDAS assessment of various models. While traditional metrics such as MRR or Hit@K identify top-ranked models, EDAS offers a deeper insight into performance consistency and robustness. For example, RotatE and NNKGC have similar MRR values, but EDAS ranks NNKGC significantly higher (Rank 3) compared to RotatE (Rank 15) over FB15k-237, highlighting NNKGC's more consistent performance. Similarly, HittER and SimKGC exhibit the same Hit@1 over FB15k-237, but EDAS ranks HittER at 20 and SimKGC at 4, showcasing differences in their overall behavior. Two models may have comparable MRR or Hit@K values but differ significantly in their weighted positive (WPav) and negative (WNav) distances from the mean. EDAS integrates these variances into WPav and WNav scores, which are normalized to compute a final M-value. The M-value synthesizes overall performance, incorporating both consistent behavior and deviations, offering a more holistic evaluation than traditional metrics alone. This demonstrates that EDAS provides a nuanced and comprehensive assessment of model performance, highlighting aspects often overlooked by conventional measures.

Table 7: The comparison of proposed and existing methods using EDAS assessment across all datasets used in the paper. Where WPav and WNav represent weighted positive and negative distances, respectively. The normalized values are denoted as N(WPav) and N(WNav), while M represents the final score and Ranks denote the rank based on M.

| Model | WPav | WNav | NWPav | NWNav | M | Rank |
|---|---|---|---|---|---|---|
| RESCAL (Nickel et al., 2011) | 0.0121 | 0.0543 | 0.0455 | 0.0543 | 0.4956 | 14 |
| TransE (Bordes et al., 2013) | 0.0029 | 0.2113 | 0.0110 | 0.2113 | 0.3999 | 16 |
| TuckER (Wang et al., 2019) | 0.0181 | 0.0306 | 0.0677 | 0.0306 | 0.5186 | 11 |
| ComplEx (Trouillon et al., 2016) | 0.0110 | 0.2557 | 0.0413 | 0.2557 | 0.3928 | 17 |
| ConvE (Vashishth et al., 2020) | 0.0514 | 0.1000 | 0.1925 | 0.1000 | 0.5462 | 10 |
| DistMult (Yang et al., 2014) | 0 | 0.2321 | 0 | 0.2321 | 0.3839 | 18 |
| RotatE (Sun et al., 2019) | 0.0045 | 0.0444 | 0.0169 | 0.0444 | 0.4862 | 15 |
| CompGCN (Vashishth et al., 2020) | 0.0154 | 0.0321 | 0.0575 | 0.0321 | 0.5127 | 12 |
| HittER (Chen et al., 2021) | 0.0080 | 1 | 0.0298 | 1 | 0.0149 | 20 |
| HAKE (Zhang et al., 2020a) | 0.0108 | 0.0196 | 0.0406 | 0.0196 | 0.5105 | 13 |
| BiQUE (Liu et al., 2024) | 0.0925 | 0.0707 | 0.3464 | 0.0707 | 0.6378 | 7 |
| StAR (Wang et al., 2021) | 0.0066 | 0.2396 | 0.0246 | 0.2396 | 0.3925 | 19 |
| MEM-KGC (w/o EP) (Choi et al., 2021) | 0.0273 | 0.0087 | 0.1023 | 0.0087 | 0.5468 | 9 |
| MEM-KGC (w/ EP) (Choi et al., 2021) | 0.0595 | 0.0031 | 0.2229 | 0.0031 | 0.6099 | 8 |
| SimKGC (Wang et al., 2022) | 0.1650 | 0.0163 | 0.6182 | 0.0163 | 0.8009 | 4 |
| DIFT (Liu et al., 2024) | 0.2363 | 0 | 0.8852 | 0 | 0.9426 | 2 |
| KICGPT (Wei et al., 2023) | 0.1247 | 0 | 0.4671 | 0 | 0.7335 | 6 |
| NBFNet (Zhu et al., 2021) | 0.1450 | 0 | 0.5432 | 0 | 0.7716 | 5 |
| NNKGC (Li & Yang, 2023) | 0.1659 | 0.0099 | 0.6214 | 0.0099 | 0.8058 | 3 |
| CAB-KGC (Liu et al., 2024) | 0.2670 | 0.0072 | 1 | 0.0072 | 0.9964 | 1 |

# B    SUPPLEMENTARY ASSESSMENT OF CAB-KGC ON SMALL DATASETS

We additionally evaluated the proposed CAB-KGC model on various small datasets, including NATION, LOCATION, SPORT, and UML datasets. The GitHub repository Awesome-Knowledge-Graph-Reasoning [2] provides a comprehensive collection of freely available datasets for Knowledge Graph Completion (KGC) and reasoning tasks. This repository is a valuable resource for researchers and practitioners, offering well-curated datasets to facilitate the evaluation and benchmarking of knowledge graph reasoning models. To further validate CAB-KGC, a detailed experimental study was conducted using metrics such as MRR, Hit@1, and Hit@3 on these datasets, comparing different variations of CAB-KGC. As shown in Table 8, the results indicate that CAB-KGC consistently outperforms other variations. These findings demonstrate that integrating knowledge graphs, neighboring entities, and relationship information significantly enhances the performance of KGC. Additionally, the evaluation mechanism EDAS, developed in this study, was utilized to assess the proposed model, other LLMs, and CAB-KGC variations across datasets like nations, locations, sports, and UMLs. Detailed experimental results confirm that CAB-KGC effectively, offering a more robust and accurate solution for KGC tail prediction.

Table 8: Comparison of the Proposed CAB-KGC and other Pre-trained models across various KG datasets

| Dataset | Models | MRR ↑ | Hit@1 ↑ | Hit@3 ↑ | Ranks |
|---|---|---|---|---|---|
| Nation | BERT_Seq | 0.266 | 0.0796 | 0.3215 | 3 |
| | RoBERT | 0.3101 | 0.1244 | 0.3214 | 2 |
| | DistilBERT | 0.2713 | 0.0746 | 0.3025 | 6 |
| | CAB-KGC-H_c | 0.3211 | 0.1493 | **0.3333** | 4 |
| | CAB-KGC-R_c | 0.3179 | 0.1493 | **0.3333** | 5 |
| | Proposed CAB-KGC | **0.3326** | **0.1642** | **0.3333** | **1** |
| Location | BERT_Seq | 0.1156 | 0.0769 | 0.1231 | 5 |
| | RoBERT | 0.5018 | 0.4154 | 0.5692 | 4 |
| | DistilBERT | 0.5394 | 0.4601 | 0.5846 | 3 |
| | CAB-KGC-H_c | 0.5375 | 0.4308 | 0.6154 | 2 |
| | CAB-KGC-R_c | 0.1249 | 0.0769 | 0.1385 | 6 |
| | Proposed CAB-KGC | **0.5788** | **0.4615** | **0.6769** | **1** |
| Sport | BERT_Seq | 0.4437 | 0.1536 | 0.7039 | 4 |
| | RoBERT | 0.4767 | 0.1704 | **0.9944** | 2 |
| | DistilBERT | 0.4509 | 0.1441 | 0.9413 | 3 |
| | CAB-KGC-H_c | 0.5004 | 0.1307 | 0.7263 | 1 |
| | CAB-KGC-R_c | 0.4255 | 0.1397 | 0.5196 | 5 |
| | Proposed CAB-KGC | **0.5319** | **0.2263** | 0.9749 | **1** |
| UMLs | BERT_Seq | 0.2537 | 0.0923 | 0.2738 | 2 |
| | RoBERT | 0.207 | 0.1044 | 0.2753 | 5 |
| | DistilBERT | 0.256 | 0.1009 | 0.2693 | 3 |
| | CAB-KGC-H_c | 0.2313 | 0.0989 | 0.2265 | 4 |
| | CAB-KGC-R_c | 0.2287 | 0.0947 | 0.2482 | 6 |
| | Proposed CAB-KGC | **0.2687** | **0.1132** | **0.2809** | **1** |

---

[2]https://github.com/LIANGKE23/Awesome-Knowledge-Graph-Reasoning

