# OpenReview forum: "CAB-KGC: Context-Aware BERT for Knowledge Graph Completion"
_ICLR.cc/2025/Conference — ICLR 2025 Conference Withdrawn Submission_

### Official Review · Reviewer_dGcv · 2024-10-19

**Soundness:** 1
**Presentation:** 2
**Contribution:** 1
**Rating:** 3
**Confidence:** 5

**Summary:**

This paper proposes a text-based KGC method named CAB-KGC. CAB-KGC finetunes a BERT model to complete the missing entity of a triple with the help of neighboring contexts, instead of entity descriptions. The proposed method gets rid of the high computational complexity imposed by negative sampling.

**Strengths:**

1) The problem is clearly defined.
2) The limitation of text-based methods, namely encoding negative samples, is clearly pointed out.
3) The proposed method can alleviate the high computational overhead imposed by negative sampling.

**Weaknesses:**

1) On page 2, line 2-7, The authors classify related works using BERT-based pre-trained language models, such as KG-BERT. This reviewer does not agree with such a claim to some extent. Given the limited number of parameters, BERT, especially BERT-base, cannot be considered as an LLM. In this case, this reviewer suggests the authors use "text-based" methods instead.
2) There are several non-concurrent state-of-the-art works that are not discussed or compared with the proposed method.

a. For instance, the SOTA GNN-based method, NBF-Net [1], accepted to NeurIPS 2021, achieves Hits@1 performance of 0.321 and 0.497 on the FB15k237 and WN18RR datasets, respectively.

b. In addition, the authors did not completely report the performance of KICGPT [2] (Wei et al. 2024) referred in their paper. KICGPT is accepted to EMNLP2023 Findings, which achieves a Hits@1 of 0.327 and an MRR of 0.412 on the FB15k237 dataset, which outperforms the proposed method.

Given the reasons above, it is inappropriate to claim that the proposed method is SOTA.

3) This reviewer acknowledges that text-based methods underperform SOTA also have their merits. However, the performance of baseline methods should be completely referred and duly acknowledged.

4) Moreover, there is another SOTA method DIFT [3], which the authors claimed that the paper is accepted to 2024, achieves a Hits@1 of 0.364 and a Hits@3 of 0.439 on the FB15k237 dataset. Nevertheless, The decision made by this reviewer is not based on the existence of [3] since it is a concurrent work of this submission.

5) This paper does not report the Hits@10 performance of the proposed method and its baselines. Hits@10 is also a commonly used evaluation metric. It is acceptable if the proposed method cannot achieve desirable Hits@10 performance, as long as the issue is discussed and analysed.

6) There are many duplicated references included in the reference list. Papers including but not limited to "Convolutional 2d knowledge graph embeddings" and KICGPT appear twice or even three times on pages 9-11.

7) There are several minor Latex formatting issues. E.g. “Equations 1-??” on page 4. The correct symbol to represent membership is \\in, not \\epsilon (the misuses are shown in the 2nd paragraph, section 3, page 4).

[1] Neural Bellman-Ford Networks: A General Graph Neural Network Framework for Link Prediction (NeurIPS 2021)

[2] KICGPT: Large Language Model with Knowledge in Context for Knowledge Graph Completion (EMNLP2023 Findings)

[3] Finetuning Generative Large Language Models with Discrimination Instructions for Knowledge Graph Completion (on Arxiv, the authors of the paper claimed that it is accepted to ISWC2024)

**Questions:**

This reviewer sincerely requests the authors to revise the manuscript carefully.

For details, please see the weakness part above.

**Details Of Ethics Concerns:**

The authors selectively report evaluation results of SOTA method KICGPT [2] in order to claim the proposed method as SOTA.

[2] KICGPT: Large Language Model with Knowledge in Context for Knowledge Graph Completion (EMNLP2023 Findings)

---

> ### Author Response · Authors · 2024-11-26
> **Response to the Reviewer**
>
> **Strengths:**
>
> We are glad the reveiwer finds our model good at alleviating computational overheads in KGC.
>
> **Weaknesses:**
>
> 1. _On page 2, line 2-7, The authors classify related works using BERT-based pre-trained language models, such as KG-BERT. This reviewer does not agree with such a claim to some extent. Given the limited number of parameters, BERT, especially BERT-base, cannot be considered as an LLM. In this case, this reviewer suggests the authors use "text-based" methods instead._
>
> $\textcolor{blue}{Response:}$ We agree with the reveiwer that models like KG-BERT and ours are better classifed as text-based rather than LLM-based models. We have revised the classification in the manuscript to reflect this.
>
> 2. _There are several non-concurrent state-of-the-art works that are not discussed or compared with the proposed method._
> _a. For instance, the SOTA GNN-based method, NBF-Net [1], accepted to NeurIPS 2021, achieves Hits@1 performance of 0.321 and 0.497 on the FB15k237 and WN18RR datasets, respectively._
>
> $\textcolor{blue}{Response:}$ Thank you for your suggestion. We have now included a comparison with the mentioned method, NBF-Net [1], in the manuscript, specifically referring to its performance on the FB15k237 and WN18RR datasets.
>
> _b. In addition, the authors did not completely report the performance of KICGPT [2] (Wei et al. 2024) referred in their paper. KICGPT is accepted to EMNLP2023 Findings, which achieves a Hits@1 of 0.327 and an MRR of 0.412 on the FB15k237 dataset, which outperforms the proposed method._
>
> _Given the reasons above, it is inappropriate to claim that the proposed method is SOTA._
> 3. _This reviewer acknowledges that text-based methods underperform SOTA also have their merits. However, the performance of baseline methods should be completely referred and duly acknowledged._
>
> $\textcolor{blue}{Response:}$ We apologize for the oversight in not including some of the evaluation metrics from the KICGPT [2] paper. This omission was unintentional and not meant to selectively report results. As the reviewer correctly points out, KICGPT outperforms our model on the FB15k237 dataset in terms of Hits@1 and MRR. However, our model performs better on the WN18RR dataset. Additionally, when considering the EDAS ranking across all datasets, our model ranks first, while KICGPT ranks 6th. Moreover, our model outperforms other state-of-the-art methods on the CoDEx-S and ConceptNet100K datasets, for which we were unable to locate KICGPT performance data in the literature. We have updated the manuscript to ensure a more comprehensive and transparent comparison. Based on these comparisons, we conclude that our model achieves state-of-the-art performance on three out of the four datasets.  We have updated the manuscript to ensure a more comprehensive and transparent comparison and proper acknowledgement.
>
> 4. _Moreover, there is another SOTA method DIFT [3], which the authors claimed that the paper is accepted to 2024, achieves a Hits@1 of 0.364 and a Hits@3 of 0.439 on the FB15k237 dataset. Nevertheless, The decision made by this reviewer is not based on the existence of [3] since it is a concurrent work of this submission._
>
> $\textcolor{blue}{Response:}$ Thank you for pointing this out. We have now included the DIFT [3] method in our paper, which is out just this month. According to our EDAS rankings, DIFT ranks number 2 when taking the performance values across datasets together. We have updated Tables 3 and 5 to reflect this addition and provide a more complete comparison.
>
> 5. _This paper does not report the Hits@10 performance of the proposed method and its baselines. Hits@10 is also a commonly used evaluation metric. It is acceptable if the proposed method cannot achieve desirable Hits@10 performance, as long as the issue is discussed and analysed._
>
> $\textcolor{blue}{Response:}$ We have now calculated and reported the Hits@10 performance for the proposed method as well as for all baseline models across the datasets.
>
> 6. _There are many duplicated references included in the reference list. Papers including but not limited to "Convolutional 2d knowledge graph embeddings" and KICGPT appear twice or even three times on pages 9-11._
>
> $\textcolor{blue}{Response:}$ Thank you for pointing this out. We have carefully reviewed the reference list and removed all duplicates, ensuring that each paper is cited only once.
>
> 7. _There are several minor Latex formatting issues. E.g. “Equations 1-??” on page 4. The correct symbol to represent membership is \in, not \epsilon (the misuses are shown in the 2nd paragraph, section 3, page 4)._
> $\textcolor{blue}{Response:}$ We have corrected the reference and ensured that the proper symbol for membership is used.

---

> > ### Author Response · Authors · 2024-11-26
> > **Response to the Reviewer Continued...**
> >
> > _[1] Neural Bellman-Ford Networks: A General Graph Neural Network Framework for Link Prediction (NeurIPS 2021)_
> >
> > _[2] KICGPT: Large Language Model with Knowledge in Context for Knowledge Graph Completion (EMNLP2023 Findings)_
> >
> > _[3] Finetuning Generative Large Language Models with Discrimination Instructions for Knowledge Graph Completion (on Arxiv, the authors of the paper claimed that it is accepted to ISWC2024)_
> >
> > $\textcolor{blue}{Response:}$ We have added these references.
> >
> > **Questions:**
> > _This reviewer sincerely requests the authors to revise the manuscript carefully._
> >
> > _For details, please see the weakness part above._
> >
> > _Flag For Ethics Review: Yes, Other reasons (please specify below)
> > Details Of Ethics Concerns:
> > The authors selectively report evaluation results of SOTA method KICGPT [2] in order to claim the proposed method as SOTA._
> >
> > _[2] KICGPT: Large Language Model with Knowledge in Context for Knowledge Graph Completion (EMNLP2023 Findings)_
> >
> > $\textcolor{blue}{Response:}$ Thank you for raising this concern. Again, we apologize for the oversight in not including some of the evaluation metrics from the KICGPT [2] paper. This was an unintentional omission and not an attempt to selectively report results. We have corrected the issue by adding the missing metrics and ensure that the comparison is complete and transparent. Our goal is to present a fair and objective evaluation of the proposed method in relation to the state-of-the-art approaches.As stated above, our model achieves state-of-the-art performance on three out of the four datasets.

---

> ### Comment · Reviewer_dGcv · 2024-11-27
> **Reply to the authors' Rebuttal.**
>
> This reviewer sincerely acknowledges the authors' work in improving the manuscript. Despite having some merits, this reviewer believes that the current version still clearly left behind the acceptance threshold of ICLR2025 from the following four perspectives.
>
> 1) Novelty
>
> Including contextual information of relations and neighboring entities for the encoding of central entities is not a brand-new idea but a natural and usual operation in KG literature [1,2,3,4], where [3] and [4] have also utilized textual semantics. [1-4] also utilize classification loss to effectively get rid of negative sampling in KG entity typing, a subtask of KGC.
>
> 2) Significance
>
> The proposed method is claimed to address the bottlenecks of text-based KGC tasks brought by negative sampling.  Nevertheless, the solution to the negative sampling has been proposed in ACL 2022 [5]. The proposed method fails to outperform SimKGC [5] on WN18RR in terms of Hits@3 and Hits@10, demonstrating its limited generality. In addition, the authors did not compare SimKGC on CoDEx-S and ConceptNet-100k. **The lack of comparison between similar solutions on different datasets further weakened the soundness of the main claim and the main contribution**.
>
> 3) Technical Quality
>
> a) Some of the claims made in this paper are insufficiently sound. Previously, this reviewer mentioned that the authors falsely claimed that the proposed method achieved state-of-the-art performance on the FB15k237 dataset. This version still exhibits similar cases on other datasets. For instance, CAB-KGC significantly underperforms SimKGC in terms of Hits@3 and Hits@10 on WN18RR. **In addition, on the **CN100K** dataset the proposed CAB-KGC method left largely behind unrevealed baseline methods [6] and [7] published in AAAI2020 and ACL2021, respectively (0.253 vs 0.3942 and 0.452 in terms of Hits@1).** This indicates **the lack of adequate discussions and comparisons between related works and the proposed method**. This reviewer would also like to emphasize that, whether intentional or not, the omission of such a large amount of baseline experimental results is unacceptable.
>
> b) The proposed method achieves **unsatisfactory Hits@10 performance** on the two commonly adopted datasets. The current version *still lacks sufficient discussion on the root causes*. Moreover, instead of reporting numbers, the text part in section 3.2 should include a detailed analysis.
>
> 4) Clarity
>
> The formatting violates the guidelines stated in the Call of Papers webpage of ICLR2025. Adjusting the margins in the style file is not allowed. This reviewer suggests that the authors move supplementary contents into appendices.
>
> Given the reasons mentioned above, this reviewer has decided to maintain the rejection rating.
>
> [1] Pan et al., Context-aware Entity Typing in Knowledge Graphs (EMNLP 2021)
>
> [2] Jin et al., A Good Neighbor, A Found Treasure: Mining Treasured Neighbors for Knowledge Graph Entity Typing (EMNLP2023)
>
> [3] Hu et al., Transformer-based Entity Typing in Knowledge Graphs (EMNLP2023)
>
> [4] Li et al., The Integration of Semantic and Structural Knowledge in Knowledge Graph Entity Typing (NAACL2024)
>
> [5] Wang et al., SimKGC: Simple Contrastive Knowledge Graph Completion with Pre-trained Language Models (ACL2022)
>
> [6] Malaviya et al., Commonsense Knowledge Base Completion with Structural and Semantic Context (AAAI2020)
>
> [7] Lovelace et al., Robust Knowledge Graph Completion with Stacked Convolutions and a Student Re-Ranking Network (ACL2021)

---

### Official Review · Reviewer_K5Ug · 2024-10-24

**Soundness:** 2
**Presentation:** 1
**Contribution:** 2
**Rating:** 3
**Confidence:** 5

**Summary:**

This paper introduces the CAB-KGC (Context-Aware BERT for Knowledge Graph Completion) model, which leverages contextual information from neighboring entities and relationships to predict tail entities, thus eliminating the reliance on entity descriptions. The model uses MLE to build loss, rather than contrastive training that requires negative sampling, which improves computational efficiency.

Additionally, the paper evaluates model performance with an additional metric, the Evaluation based on Distance from Average Solution (EDAS), for more comprehensive assessment. Through experiments on the FB15k-237 and WN18RR datasets, CAB-KGC outperforms some baseline methods, showing improvements in metrics like Hit@1.

**Strengths:**

1. The CAB-KGC model presents a novel approach by leveraging the contextual information of neighboring entities and relationships without relying on entity descriptions or negative triplet sampling, which is a common limitation in previous KGE and LLM-based methods. This removes the dependency on external textual information, making it applicable to a wider variety of KGs, especially those that lack entity descriptions. This design leads to more efficient training and improved evaluation performance.

2. The paper demonstrates thorough experimentation and validation of the proposed CAB-KGC model on standard benchmark datasets (FB15k-237 and WN18RR).

3. The introduction of the EDAS criterion also has the potential to influence future performance evaluation practices in the knowledge graph domain.

**Weaknesses:**

1. Lack of novelty: The innovation in this work seems incremental, as it mainly builds on the SimKGC framework. The only major difference in the CAB-KGC model is that it does not require head entity descriptions and employs a classification loss (cross-entropy) instead of contrastive loss for training.
2. Presentation issues:

    2.1. Unclear figures: Figures are often unclear or poorly labeled, making it hard for readers to interpret their meaning. For instance, Figure 1 lacks detailed labeling and a proper explanation of how its components relate to the proposed methodology.

    2.2. Inconsistent mathematical notation: Symbols are used inconsistently. In the Introduction, the sets of entities and relations are referred to as $\mathcal{E}$ and $\mathcal{R}$, but in the Methodology section, they are denoted as $E$ and $R$. Additionally, the formulas presented lack rigor and are not sufficiently academic.

    2.3. Grammatical and typographical errors: The paper contains several issues with grammar and typos.

    2.4. Missing ablation studies: Ablation results and analyses are absent from the main text, and since reviewers are not required to consult the appendix, this omission is problematic. The authors should revise the structure of the paper.

3. Experimental design shortcomings:

    3.1. Small datasets: The experiments are conducted primarily on small datasets like WN18RR and FB15k-237. While these are common, evaluating the model on larger datasets, such as Wikidata5M, would better demonstrate the method’s generalizability.

    3.2. Training epoch limitations: The authors note that “The number of epochs was set to 30 for CAB-KGC and other models,” which may result in unfair comparisons since different models might require different numbers of training epochs.

    3.3. Lack of metric comparison: There is insufficient comparison between EDAS and traditional metrics like MRR, and the paper does not thoroughly explain the advantages of EDAS.

    3.4. No explanation of ablation results: The ablation results are not properly explained, making it difficult to assess their relevance or impact.

**Questions:**

The authors can focus on addressing the third concern of the weaknesses (Experimental design shortcomings), providing additional results, clarifications, etc.

---

> ### Author Response · Authors · 2024-11-26
> **Response to the Reviewer**
>
> **Strengths:**
>
> Thank you for recognizing the key contributions of our work, particularly the novelty of our approach expoiting context-aware information, the introduction of the EDAS metric and our comprehensive comparative experiments.
>
> **Weaknesses:**
>
> 1. _Lack of novelty: The innovation in this work seems incremental, as it mainly builds on the SimKGC framework. The only major difference in the CAB-KGC model is that it does not require head entity descriptions and employs a classification loss (cross-entropy) instead of contrastive loss for training._
>
> $\textcolor{blue}{Response:}$ Kindly see our above comment 'Clarifying the Novelty and Impact of CAB-KGC'  for a detailed response to this point. https://openreview.net/forum?id=lBrLDC7qXF&noteId=tqOv4egDUX
>
> **Presentation issues:**
>
> 2.1. _Unclear figures: Figures are often unclear or poorly labeled, making it hard for readers to interpret their meaning. For instance, Figure 1 lacks detailed labeling and a proper explanation of how its components relate to the proposed methodology._
>
> $\textcolor{blue}{Response:}$ Figure 1 has been revised to highlight all the model components clearly, with improved annotations and detailed captions to faciliate better understanding. We hope the  reveiwer finds the revsied figure clear and easy to interpret.
>
> 2.2. _Inconsistent mathematical notation: Symbols are used inconsistently. In the Introduction, the sets of entities and relations are referred to as and , but in the Methodology section, they are denoted as  and. Additionally, the formulas presented lack rigor and are not sufficiently academic._
>
> $\textcolor{blue}{Response:}$ We have ensured that all symbols and notations used in the paper are now consistent throughout and aligned with the definitions provided in Table 2, which lists all symbols and notations for clarity and reference. Additionally, we have revised the formulas for greater rigor and academic precision.
>
> 2.3. _Grammatical and typographical errors: The paper contains several issues with grammar and typos._
>
> $\textcolor{blue}{Response:}$ We have thoroughly reviewed the paper and corrected all grammatical and typographical errors to improve the clarity and readability of the paper.
>
> 2.4. _Missing ablation studies: Ablation results and analyses are absent from the main text, and since reviewers are not required to consult the appendix, this omission is problematic. The authors should revise the structure of the paper._
>
> $\textcolor{blue}{Response:}$ As suggested we have added the ablation studies into main text. The contribution of each component in CAB-KGC, including “head context” and “relationship context,” is detailed in Table 3 in these ablation studies. These ablations prove that both components are important and removal of any component decreases the model performance. We also note that relationship context has a smaller impact on model performance than the head context. This is expected as head context provides specific local information about the node while relation context gives a general semantic of relation in overall LG.  Furthermore, the relationship conext is generally larger in size than head context, thereby adding a bit of extra noise into the model, which affects performance.

---

> > ### Author Response · Authors · 2024-11-26
> > **Response to the Reviewer Continued...**
> >
> > **Experimental design shortcomings:**
> >
> > 3.1. _Small datasets: The experiments are conducted primarily on small datasets like WN18RR and FB15k-237. While these are common, evaluating the model on larger datasets, such as Wikidata5M, would better demonstrate the method's generalizability._
> >
> > $\textcolor{blue}{Response:}$ We have conducted additional testing on more datasets, specifically ConceptNet100K and the CoDEx-S an extracted version of Wikidata5M, to further evaluate the generalizability of CAB-KGC. We were not able to run it on Wikidata5M owing to our limited computational resources and revision time of this manuscript. CAB-KGC outperforms SOTA models in three of four datasets.
> >
> > 3.2. _Training epoch limitations: The authors note that “The number of epochs was set to 30 for CAB-KGC and other models,” which may result in unfair comparisons since different models might require different numbers of training epochs._
> >
> > $\textcolor{blue}{Response:}$ Thank you for highlighting. This was a misreport on our part. Training for the CAB-KGC model and its variants was halted once the evaluation metrics stabilized to the third decimal place, typically within 30 epochs. This approach ensures fair comparison by stopping training when model performance plateaus, regardless of the number of epochs required. We believe this method provides a more objective evaluation across models.
> >
> > 3.3. _Lack of metric comparison: There is insufficient comparison between EDAS and traditional metrics like MRR, and the paper does not thoroughly explain the advantages of EDAS._
> >
> > $\textcolor{blue}{Response:}$ Kindly see our justification in our above comment titled 'Justifying EDAS metric for Fair and Robust KGC Evaluation'
> >
> > 3.4. _No explanation of ablation results: The ablation results are not properly explained, making it difficult to assess their relevance or impact._
> >
> > $\textcolor{blue}{Response:}$ The explanations are provided in above answer as well as the in Results section of the paper where the ablation studies and reported and discussed.

---

> > > ### Comment · Reviewer_K5Ug · 2024-11-29
> > >
> > > The reviewer appreciates the authors’ revisions to the manuscript; however, significant concerns regarding representation, contribution, and experimental design persist. Therefore, the reviewer will keep the score.

---

### Official Review · Reviewer_e1sp · 2024-10-31

**Soundness:** 3
**Presentation:** 3
**Contribution:** 2
**Rating:** 6
**Confidence:** 3

**Summary:**

This paper introduces CAB-KGC, which eliminates the need for entity descriptions and negative triplet sampling, reducing computation while improving performance. It leverages contextual information from neighboring entities and relationships to predict tail entities in knowledge graphs.

**Strengths:**

1.The paper is clearly written and easy to follow.

2. The proposed CAB-KGC does not require negative sample training, enhancing training speed and resilience against negative sample selection, and eliminates reliance on entity descriptions, focusing solely on head and relationship contexts.

**Weaknesses:**

1. The article only uses two datasets in its experiments and lacks large-scale datasets. The authors should consider supplementing the datasets.

2. The creation of Figure 1 is evidently too rough, including misaligned text and meaningless graphics in the small image on the left.

3. The model section of this paper is very brief, with the model being simply a BERT that takes in information from neighboring nodes. While this approach may be effective, I question whether the paper's innovation and interpretability are sufficient for acceptance at ICLR.

4 The paper repeatedly emphasizes the advantages of the proposed model on effiency; therefore, it would be beneficial to include comparative experiments on time complexity or runtime performance between the proposed model and the baseline.

**Questions:**

Please refer to the "weaknesses" section.

---

> ### Author Response · Authors · 2024-11-26
> **Response to the Reviewer**
>
> **Strengths:**
>
> We are happy that the reveiwer finds the paper clearly written and our proposal as an efficient approach to KGC.
>
> **Weaknesses:**
>
> 1. _The article only uses two datasets in its experiments and lacks large-scale datasets. The authors should consider supplementing the datasets._
>
> $\textcolor{blue}{Response:}$ We have conducted additional testing on more datasets, specifically ConceptNet100K and the CoDEx-S an extracted version of Wikidata5M (the suggested dataset from one of the reveiwers), to further evaluate the generalizability of CAB-KGC. We were not able to run it on Wikidata5M owing to our limited computational resources and revision time of this manuscript. CAB-KGC outperforms SOTA models in three of four datasets.
>
> 2. _The creation of Figure 1 is evidently too rough, including misaligned text and meaningless graphics in the small image on the left._
>
> $\textcolor{blue}{Response:}$ Figure 1 has been revised to highlight all the model components clearly, with improved annotations and detailed captions to faciliate better understanding. Misaligned text and unncessary graphs has been removed as suggested.
>
>
> 3. _The model section of this paper is very brief, with the model being simply a BERT that takes in information from neighboring nodes. While this approach may be effective, I question whether the paper's innovation and interpretability are sufficient for acceptance at ICLR._
>
> $\textcolor{blue}{Response:}$ Kindly see our above comment 'Clarifying the Novelty and Impact of CAB-KGC' in response.
>
> 4. _The paper repeatedly emphasizes the advantages of the proposed model on effiency; therefore, it would be beneficial to include comparative experiments on time complexity or runtime performance between the proposed model and the baseline._
>
> $\textcolor{blue}{Response:}$ Thank you for your suggestion. While we acknowledge that comparative experiments on time complexity or runtime performance would provide further insights into the efficiency of our model, we were unable to conduct these experiments within the current revision timeframe.
> That being said, CAB-KGC is designed with computational efficiency in mind. It balances performance with computational cost by utilizing concise adjacent contextual data from the knowledge graph and reducing the input token size for BERT by omitting lengthy entity descriptions. Additionally, the model avoids the computational overhead of negative triplet sampling, further minimizing unnecessary processing. The context extraction step is lightweight and occurs only once per dataset, which significantly reduces computational effort.
> Although the primary computational expense in our model stems from BERT training (like in KG-BERT model), this cost is offset by the substantial performance improvements, as evidenced by the state-of-the-art results we report. We believe these performance gains justify the computational investment, and future work will explore more detailed comparisons with baseline models on runtime efficiency as part of a broader evaluation framework. Note that the current SOTA model DIFT uses 7B parameters while our model has only BERT's 110M parameters for training.

---

### Official Review · Reviewer_Y4mx · 2024-11-03

**Soundness:** 2
**Presentation:** 2
**Contribution:** 2
**Rating:** 3
**Confidence:** 4

**Summary:**

This paper proposes Context Aware BERT for Knowledge Graph Completion (CAB-KGC), which introduces contextual information to entities and relationships in KG, eliminates the need for entity descriptions and negative sampling, and reduces computational complexity while improving performance. In addition, this paper proposes an EDAS evaluation method to more comprehensively assess the performance of model.

**Strengths:**

S1. The paper focuses on knowledge graph completion, which is an important issue. The language model-based method discussed in this paper exhibit a certain novelty compared to traditional structure-based approaches.

S2. The Evaluation based on Distance from Average Solution (EDAS) criteria used in the paper is a relatively novel evaluation metric that can better evaluate model performance in the presence of multiple metrics.

**Weaknesses:**

W1. This work has limited technical contributions. This approach simply concatenates entities, relationships, and their contexts and inputs them into the language model, then calculates the probability of all available entities as tail entities. The idea lacks novelty.

W2. The description of methodology is not clear enough. The paper does not clearly specify the output of language model or how the language model output is used to compute the probability of the tail entity. Although Figure 2 suggests that the authors intend to use the embedding of CLS token for multi-class classification, this is not explicitly stated in the text.

W3. The paper suffers from a lack of consistency in its symbolic notation, which may lead to confusion for readers. For example, in the problem formulation paragraph of the methodology section, there are three different symbol expressions for knowledge graph G.

W4. The paper contains numerous errors in details. For example, the Hadamard product symbol in the score function of RotatE is incorrect; there is an undefined symbol g in equation (11); the citations for ChatGPTzero-shot and ChatGPTone-shot are incorrect in Table 2.

**Questions:**

How were the smaller datasets, such as NATION, LOCATION, COUNTRY, SPORT, and UML, constructed? Additionally, why were the ablation experiments conducted on these smaller datasets instead of on larger benchmarks like FB15k-237 and WN18RR?

---

> ### Author Response · Authors · 2024-11-26
> **Response to the Reviewer**
>
> **Strengths:**
>
> We are glad that the reveiwer see certain novelty in our approach as well as in the introduction of the EDAS metric for KGC.
>
> **Weaknesses:**
>
> W1. Limited Technical Contributions:
>
>  _This approach simply concatenates entities, relationships, and their contexts and inputs them into the language model, then calculates the probability of all available entities as tail entities. The idea lacks novelty._
>
> $\textcolor{blue}{Response:}$ Kindly see our above comment 'Clarifying the Novelty and Impact of CAB-KGC' in response.
>
> W2. _Unclear Methodology Description:_
>  _The paper does not clearly specify the output of the language model or how the language model output is used to compute the probability of the tail entity. Although Figure 2 suggests that the authors intend to use the embedding of the CLS token for multi-class classification, this is not explicitly stated in the text._
>
> $\textcolor{blue}{Response:}$ The methodology section has been revised to provide a clear, step-by-step explanation of the CAB-KGC approach. The reveiwer is correct to figure out that use the embedding of the CLS token for multi-class classification. We have explicity clarified this in the revised methodology.
>
> W3. _Inconsistent Symbolic Notation:_
>  _The paper suffers from a lack of consistency in its symbolic notation, which may lead to confusion for readers. For example, in the problem formulation paragraph of the methodology section, there are three different symbol expressions for knowledge graph \( G \)._
>
> $\textcolor{blue}{Response:}$ We have ensured that all symbols and notations used in the paper are now consistent and aligned with the definitions provided in Table 3, which consolidates all symbols and notations for clarity and reference
>
> W4. _Numerous Errors in Details:_
>  _For example, the Hadamard product symbol in the score function of RotatE is incorrect; there is an undefined symbol in Equation (11); and the citations for ChatGPTzero-shot and ChatGPTone-shot are incorrect in Table 2._
>
> $\textcolor{blue}{Response:}$ The formulas have been revised, and the Hadamard product symbol in the score function of RotatE has been corrected. Additionally, the undefined symbol \( g \) in Equation (11) has been addressed, and the citations for ChatGPT-zero-shot and ChatGPT-one-shot in Table 2 have been corrected.
>
> **Questions:**
>
>  _How were the smaller datasets, such as NATION, LOCATION, COUNTRY, SPORT, and UML, constructed? Additionally, why were the ablation experiments conducted on these smaller datasets instead of on larger benchmarks like FB15k-237 and WN18RR?_
>
> $\textcolor{blue}{Response:}$ All these datasets are available from a survey work by Liang et al. (2024) via their GitHub repository and can be accessed at the following link: https://shorturl.at/zTX7H. This repository is a sound resource, featuring well-organized datasets, detailed documentation, and a variety of KGC tools. The experiments on smaller datasets have been removed from the ablation study. As suggested by the reveiwer, we have provided experimental results on larger benchmark datasets (FB15k-237 and WN18RR) in the revised ablation study.

---

> > ### Comment · Reviewer_Y4mx · 2024-11-26
> >
> > Thanks for your response. Now there is no more question from my perspective.

---

### Official Review · Reviewer_AKtK · 2024-11-04

**Soundness:** 2
**Presentation:** 2
**Contribution:** 2
**Rating:** 3
**Confidence:** 4

**Summary:**

This paper introduces a novel approach for Knowledge Graph Completion (KGC) named Context-Aware BERT for Knowledge Graph Completion (CAB-KGC). The goal of CAB-KGC is to predict missing entities or relationships in knowledge graphs by leveraging contextual information. Unlike traditional embedding-based methods that struggle with unseen entities and relationships, CAB-KGC utilizes the contextual data from neighboring nodes and relationships, integrating these insights with a BERT-based architecture to enhance prediction accuracy for tail entities.
The authors also propose a new evaluation metric, Evaluation based on Distance from Average Solution (EDAS), to address potential inconsistencies in existing metrics like Mean Reciprocal Rank (MRR) and Hit@k. EDAS provides a more comprehensive assessment by considering deviations from average performance across several criteria.
CAB-KGC is evaluated against state-of-the-art KGC methods on benchmark datasets FB15k-237 and WN18RR, showing improvements in standard metrics, particularly in Hit@1 and MRR. The experimental results demonstrate that CAB-KGC outperforms baseline methods across different datasets, indicating that incorporating both head and relationship contexts into BERT can improve KGC model accuracy. The paper also includes ablation studies to validate the contributions of each component within the CAB-KGC model, showing that combining head and relationship contexts yields the best results.
CAB-KGC addresses limitations in current KGC approaches by utilizing both structural and contextual information without relying on negative sampling or entity descriptions, and the EDAS metric offers an alternative way to rank model performance comprehensively.

**Strengths:**

1.	Introduction of CAB-KGC for Knowledge Graph Completion: The paper presents a novel method, CAB-KGC, which leverages context-aware information from both head entities and relationships. This is an original approach for enhancing knowledge graph completion tasks, aiming to address limitations of existing embedding-based and LLM-based models. 2.
2.	Proposal of EDAS as a New Evaluation Metric: The introduction of the EDAS (Evaluation based on Distance from Average Solution) metric is a unique contribution. EDAS aims to offer a more comprehensive assessment by incorporating both positive and negative deviations, which could provide a more nuanced evaluation of model performance, especially on ranking tasks.
3.	Comparative Experimental Results: The paper conducts extensive experiments comparing CAB-KGC with multiple state-of-the-art methods, particularly on FB15k-237 and WN18RR datasets. The results demonstrate that CAB-KGC achieves competitive performance, with notable improvements on the Hit@1 metric.

**Weaknesses:**

1. Formatting and Layout Issues:

•	The paper has several formatting inconsistencies that detract from its professionalism. For instance, the template content for acknowledgments has not been removed from the end of the main text, which is distracting and unpolished.

•	In the appendix, there are large blank spaces between figures and tables, and the line thickness in tables is inconsistent, affecting visual uniformity.

•	The title for Figure 2 is positioned too close to the bottom margin, overlapping with the page number, which can cause readability issues. These layout errors suggest a need for a thorough review of formatting before submission.

2. Insufficient Detail in Methodology and Figures:

•	The paper’s core methodological explanation is sparse and lacks clarity, particularly in the section describing CAB-KGC. Statements such as "The CAB-KGC proposed method incorporates the importance of contextual information obtained from the head entity and the relationship and integrates with BERT" are vague and need further elaboration.

•	Figures intended to illustrate the model, especially Figure 1, have limited accompanying descriptions, which could hinder reader understanding. Additionally, some figures, particularly in the Appendix, are complex and lack explanatory text, making it challenging for readers to interpret them effectively.

3. Weak Justification for the New Evaluation Metric (EDAS):

•	While EDAS is introduced as a novel evaluation metric, the paper does not sufficiently motivate its necessity. The limitations of current evaluation standards (such as MRR and Hit@k) are only briefly mentioned, and it is unclear why EDAS would provide a significant advantage. A more detailed comparison of EDAS with traditional metrics, highlighting specific scenarios where EDAS offers clearer insights, would strengthen this contribution.

4. Limited Innovation in CAB-KGC:

•	CAB-KGC’s novelty is unclear. The model combines context-aware techniques with BERT embeddings, which, while valuable, may not be a groundbreaking innovation within the knowledge graph completion (KGC) domain. The approach might appear as an incremental improvement over existing models rather than a fundamentally new technique. It would be beneficial for the authors to clarify CAB-KGC’s unique contributions and differentiate it from similar methods.

5. Insufficient Experimental Validation:

•	The paper’s experiments are conducted on only two datasets, FB15k-237 and WN18RR, which limits the demonstration of the model’s generalizability. Adding results from a dataset with entity descriptions would provide a more comprehensive evaluation and demonstrate the model’s adaptability across diverse knowledge graphs.

•	The experimental setup also lacks an ablation study on a wider set of datasets or with a broader variation of components, such as testing CAB-KGC in contexts with different entity types and relationship structures.

6. Imbalance in Content Focus:

•	Although CAB-KGC is the main proposed method, much of the paper focuses on EDAS, which may dilute the impact of CAB-KGC. A better balance could be achieved by providing more detailed insights into CAB-KGC’s architecture, implementation, and performance analysis. Readers might expect the majority of the paper to focus on the core model rather than the evaluation metric.

7. Page Length and Presentation Issues:

•	The main content is slightly shorter than the ICLR recommended 9-page limit, which might suggest a lack of comprehensive analysis or additional experimental validation. Expanding sections on methodology and experiments could provide a fuller picture of the model’s contributions.

•	In terms of presentation, the writing style sometimes lacks precision, and the images could benefit from clearer, more thorough captions and descriptions. Improved organization and clarity would enhance readability and comprehension.

**Questions:**

1. Formatting: There are formatting issues, such as the residual template text and inconsistent spacing. Will you address these for better readability?
2. EDAS Motivation: Can you elaborate on the limitations of traditional metrics that EDAS addresses, specifically in the context of KGC?
3. Methodology Clarity: The methodology has vague phrases like “incorporates contextual information.” Could you clarify this with more detail?
4. Broader Validation: Why were only two datasets used, and do you plan to test CAB-KGC’s generalizability on more datasets?
5. Figure Explanations: Could you add clearer captions or appendix notes to explain complex figures like Figure 1?
6. Ablation Analysis: Can you discuss the contribution of each component in CAB-KGC, specifically “head context” and “relationship context”?
7. Computational Cost: CAB-KGC is computationally intensive. Could you clarify the trade-offs between this cost and performance gains?

---

> ### Author Response · Authors · 2024-11-26
> **Response to the Reviewer**
>
> **Strengths:**
>
> Thank you for recognizing the key contributions of our work, particularly the novelty of our approach using context-aware information, the introduction of the EDAS metric and our comprehensive comparative experiments.
>
> **Weaknesses:**
>
> 1. _Formatting and Layout Issues:_
> _The paper has several formatting inconsistencies that detract from its professionalism. For instance, the template content for acknowledgments has not been removed from the end of the main text, which is distracting and unpolished_
> _In the appendix, there are large blank spaces between figures and tables, and the line thickness in tables is inconsistent, affecting visual uniformity._
>  _The title for Figure 2 is positioned too close to the bottom margin, overlapping with the page number, which can cause readability issues. These layout errors suggest a need for a thorough review of formatting before submission._
>
> $\textcolor{blue}{Response:}$ We have addressed the formatting issues as follows: We removed the placeholder acknowledgment content and corrected formatting inconsistencies for a more polished and professional presentation. We removed the large blank spaces between tables in the appendix and ensured consistent line thickness in the tables. Figure 2 has been removed as it conveyed redundant information already covered by Figure 1.
>
> 2. _Insufficient Detail in Methodology and Figures:_
> * _The paper's core methodological explanation is sparse and lacks clarity, particularly in the section describing CAB-KGC. Statements such as "The CAB-KGC proposed method incorporates the importance of contextual information obtained from the head entity and the relationship and integrates with BERT" are vague and need further elaboration._
>
> $\textcolor{blue}{Response:}$  The methodology section has been thoroughly revised to provide a step-by-step explanation of the CAB-KGC approach, ensuring clarity and specificity. These updates are reflected on pages 4, 5, and 6 of the revised manuscript. The statement in question has been refined to, -- "To achieve accurate tail prediction, the proposed CAB-KGC method extracts contextual information from the knowledge graph surrounding the head entity and the relationship in question, and leverages this context within a BERT model to learn and generate precise predictions." This revised statement is now followed by a comprehensive description of the types of contextual information extracted, the extraction process, and how this information is fed into the BERT model. For further details, please refer to the updated methodology section in the revised manuscript.
>
> * _Figures intended to illustrate the model, especially Figure 1, have limited accompanying descriptions, which could hinder reader understanding. Additionally, some figures, particularly in the appendix, are complex and lack explanatory text, making it challenging for readers to interpret them effectively._
>
> $\textcolor{blue}{Response:}$ We have revised Figure 1 to offer a clearer and more comprehensive depiction of the methodology, making it easier for readers to understand the key components and workflow of the CAB-KGC model. Additionally, Figure 2 has been removed as it conveyed redundant information already covered by Figure 1. Enhancements to Figure 1 include additional explanatory text and annotations, ensuring that readers can effectively interpret the information presented. The updated figure and its detailed description are now included on page 5 of the revised manuscript, addressing the concerns raised.
>
> 3. _Weak Justification for the New Evaluation Metric (EDAS):_
>  _While EDAS is introduced as a novel evaluation metric, the paper does not sufficiently motivate its necessity. The limitations of current evaluation standards (such as MRR and Hit@k) are only briefly mentioned, and it is unclear why EDAS would provide a significant advantage. A more detailed comparison of EDAS with traditional metrics, highlighting specific scenarios where EDAS offers clearer insights, would strengthen this contribution._
>
> $\textcolor{blue}{Response:}$ Kindly see above comment 'Justifying EDAS metric for Fair and Robust KGC Evaluation' https://openreview.net/forum?id=lBrLDC7qXF&noteId=RvCk5KSqcs for a thorough response to this point.
>
> 4. _Limited Innovation in CAB-KGC:_
> _CAB-KGC's novelty is unclear. The model combines context-aware techniques with BERT embeddings, which, while valuable, may not be a groundbreaking innovation within the knowledge graph completion (KGC) domain. The approach might appear as an incremental improvement over existing models rather than a fundamentally new technique. It would be beneficial for the authors to clarify CAB-KGC's unique contributions and differentiate it from similar methods._
>
> $\textcolor{blue}{Response:}$ Kindly see our above comment 'Clarifying the Novelty and Impact of CAB-KGC'  for a detailed response to this point. https://openreview.net/forum?id=lBrLDC7qXF&noteId=tqOv4egDUX

---

> ### Author Response · Authors · 2024-11-26
> **Response to Reviewer Continued..**
>
> 5. _Insufficient Experimental Validation:_
> _The paper's experiments are conducted on only two datasets, FB15k-237 and WN18RR, which limits the demonstration of the model’s generalizability. Adding results from a dataset with entity descriptions would provide a more comprehensive evaluation and demonstrate the model's adaptability across diverse knowledge graphs._
>
> $\textcolor{blue}{Response:}$ To address this, we expanded our experimental evaluation to include two additional datasets: ConceptNet100K and CoDEx-S (an extracted subset of Wikidata5M, as suggested by another reviewer). These datasets were selected to test the model's generalizability and adaptability across diverse knowledge graphs.
>
> While we were unable to test on the full Wikidata5M due to limited computational resources and the limited time available for revision, our results demonstrate the robustness of CAB-KGC. Specifically:
>
> 1. CAB-KGC outperforms state-of-the-art (SOTA) models on three of the four datasets,  CoDEx-S, WN18RR, and ConceptNet100K.
> 2. Although FB15k-237, WN18RR, and CoDEx-S include entity descriptions, our model does not rely on these descriptions to operate, setting it apart from many existing approaches.
> 3. On ConceptNet100K, a dataset that lacks entity descriptions, CAB-KGC  achieves superior performance compared to SOTA methods, highlighting its adaptability and independence from entity descriptions.
> These results reinforce the claim that CAB-KGC generalizes well across diverse datasets, regardless of whether entity descriptions are available.
>
> _The experimental setup also lacks an ablation study on a wider set of datasets or with a broader variation of components, such as testing CAB-KGC in contexts with different entity types and relationship structures._
>
> $\textcolor{blue}{Response:}$ As suggested, we conducted ablation studies using two datasets, FB15k-237 and WN18RR, to analyze the effects of removing key model components such as head context and relationship context. The results demonstrate:
> that removing this component resulted in the most pronounced drop in performance across both datasets, underscoring its importance. While relationship context also improves performance, its impact is relatively smaller compared to head context. This is expected since relationship context provides a broader semantic understanding of the relation's role in the global graph, which complements but does not replace the localized insights provided by head context. The smaller impact of relationship context can also be attributed to its larger size compared to head context, which may introduce an extra bit of noise into the model, slightly affecting performance. However, it still plays a crucial role in improving generalization by offering a global perspective on the relational patterns in a KG.
>
> 6. _Imbalance in Content Focus:_
> _Although CAB-KGC is the main proposed method, much of the paper focuses on EDAS, which may dilute the impact of CAB-KGC. A better balance could be achieved by providing more detailed insights into CAB-KGC's architecture, implementation, and performance analysis. Readers might expect the majority of the paper to focus on the core model rather than the evaluation metric._
>
> $\textcolor{blue}{Response:}$ We have made the following revisions to ensure a better balance:
>
> First, we have added a detailed explanation of CAB-KGC's architecture, including the logic and mechanisms for extracting and integrating head context and relationship context into the model training pipeline, in Section 3. Additionally, Figure 1 has been updated to provide a clear and comprehensive depiction of the model's workflow.
>
> Second, a subsection in the methodology now elaborates on the implementation details, including hyperparameters, dataset usage, and evaluation process, illustrating how CAB-KGC operates and can be reproduced.
> Third, we now provide a thorough discussion of CAB-KGC's SOTA performance across four datasets in the results section.
> Fourth, we added new ablation studies that demonstrate the importance of the model's components.
>
> While EDAS remains an important contribution, we have streamlined it as a secondary focus. The description of EDAS  is now consolidated into a single subsection (under Section 3.1). This ensures that CAB-KGC remains the central theme of the paper.
>
> 7. _Page Length and Presentation Issues:_
> _• The main content is slightly shorter than the ICLR recommended 9-page limit, which might suggest a lack of comprehensive analysis or additional experimental validation. Expanding sections on methodology and experiments could provide a fuller picture of the model's contributions._
>
> $\textcolor{blue}{Response:}$ We have expanded the methodology and experimental sections to provide a more comprehensive analysis and additional validation of CAB-KGC with the updated paper now spanning close to 10 pages.

---

> ### Author Response · Authors · 2024-11-26
> **Response to the Reviewer Continued ...**
>
> _• In terms of presentation, the writing style sometimes lacks precision, and the images could benefit from clearer, more thorough captions and descriptions. Improved organization and clarity would enhance readability and comprehension._
>
> $\textcolor{blue}{Response:}$ We have revised the manuscript to improve precision in writing and ensure concise descriptions and detailed captions. The overall organization has been enhanced to improve readability and comprehension.
>
> **Questions:**
>
> 1. _Formatting: There are formatting issues, such as the residual template text and inconsistent spacing. Will you address these for better readability?_
>
> $\textcolor{blue}{Response:}$ We have thoroughly revised the manuscript's formatting, ensuring that all residual template text has been removed and that spacing and inconsistencies have been carefully corrected. These changes enhance the overall readability and presentation quality of the paper.
>
> 2. _EDAS Motivation: Can you elaborate on the limitations of traditional metrics that EDAS addresses, specifically in the context of KGC?_
>
> $\textcolor{blue}{Response:}$ As discussed above, The first paragraph on metrics' limitations is added to the paper Introduction. A comparative analysis of traditional evaluation metrics and EDAS is provided in sectoin 3.1 last paragraph (page 8), highlighting how EDAS addresses its limitations in the context of KGC. Additional details and utility of EDAS is directed in Appendix A where different EDAS computations on our experimental results are discussed. Appendix A also provides a clear explanation to help readers understand how EDAS operates.
>
> 3. _Methodology Clarity: The methodology has vague phrases like “incorporates contextual information.” Could you clarify this in more detail?_
>
> $\textcolor{blue}{Response:}$ We have thoroughly revised the methodology section and removed any ambiguous terms (such as the one highlighted) to ensure that the methodology is clearly defined and understandable for the reader.
>
> 4. _Broader Validation: Why were only two datasets used, and do you plan to test CAB-KGC's generalizability on more datasets?_
>
> $\textcolor{blue}{Response:}$ We have conducted additional testing on more datasets, specifically ConceptNet100K and the CoDEx-S an extracted version of Wikidata5M (the suggested dataset from one of the reveiwers), to further evaluate the generalizability of CAB-KGC. We were not able to run it on Wikidata5M owing to our limited computational resources and revision time of this manuscript. CAB-KGC outperforms SOTA models in three of four datasets.
>
> 5. _Figure Explanations: Could you add clearer captions or appendix notes to explain complex figures like Figure 1?_
>
> $\textcolor{blue}{Response:}$ Figure 1 has been revised to highlight all the model components, with improved annotations and detailed captions to faciliate better understanding.
>
> 6. _Ablation Analysis: Can you discuss the contribution of each component in CAB-KGC, specifically “head context” and “relationship context”?_
>
> $\textcolor{blue}{Response:}$  The contribution of each component in CAB-KGC, including “head context” and “relationship context,” is detailed in Table 3 in these ablation studies. These ablations prove that both components are important and removal of any component decreases the model performance. We also note that relationship context has a smaller impact on model performance than the head context. This is expected as head context provides specific local information about the node while relation context gives a general semantic of relation in overall LG.  Furthermore, the relationship conext is generally larger in size than head context, thereby adding a bit of extra noise into the model, which affects performance.
>
> 7. _Computational Cost: CAB-KGC is computationally intensive. Could you clarify the trade-offs between this cost and performance gains?_
>
> $\textcolor{blue}{Response:}$ CAB-KGC balances computational cost and performance by utilizing concise adjacent contextual data from the knowledge graph, reducing input token size for BERT by ignoring long entity descriptions and avoiding the need for negative triplet sampling. Context extraction is lightweight and occurs once per dataset, minimizing computational effort. The main computational expense comes from BERT training, but this is justified by significant performance gains, as demonstrated by state-of-the-art results.  Kindly note that the current SOTA model DIFT uses 7B parameters while our model has only BERT's 110M parameters for training.

---

### Author Response · Authors · 2024-11-26
**Revised Manuscript and A list of changes made**

Dear Reviewers

We would like to sincerely thank you for a very thorough evaluation of our submission and your constructive feedback. Your thoughtful suggestions have significantly contributed to improving the clarity and quality of our paper. We have carefully addressed each of the points raised and made the necessary revisions to enhance the presentation and strengthen the content of our work.
We greatly appreciate the valuable time and effort you put in reviewing our manuscript, and we believe that the revisions made will help in making the paper more polished and comprehensive. Thank you again for your valuable input. We trust that the revisions adequately address the reviewer’s concerns and contribute to enhancing the overall quality and clarity of our work. We look forward to the reviewer's final evaluation based on these updates.


__Below are the key revisions made to the manuscript.__

1. __Introduction:__ Included a new paragraph addressing the limitations of commonly used assessment metrics in KGC.
2. __Table 1:__ Updated to highlight the distinctive structural features of knowledge graphs that our model leverages.
3. __Methodology and Figure 1:__ Revised thoroughly to provide a detailed explanation of our model's working.
4. __Datasets:__ Expanded experiments to include two additional datasets, CoDEx-S and ConceptNet100K.
5. __Experiments Section:__ Added a new subsection discussing the advantages of the EDAS assessment framework.
6. __Metrics Reporting:__ Provided comprehensive results for common metrics across four datasets: FB15k-237, WN18RR, CoDEx-S, and ConceptNet100K.
7. __Ablation Studies:__ Introduced a new section analyzing the contributions of our model’s components.
8. __EDAS Rankings:__ Added a table presenting the EDAS-based rankings of all evaluated models.

_The code will be shared upon the acceptance of the paper to ensure proper access and usage_.

---

### Author Response · Authors · 2024-11-26
**Clarifying the Novelty and Impact of CAB-KGC**

We appreciate the reviewers' feedback and would like to address the concerns regarding the novelty and technical contributions of our work. While our model builds on prior research, its contributions are substantial and address critical limitations in the state-of-the-art methods. Our proposal is simple but meaningful and powerful. We want to explicity exploit multiple structural features of knowledge graphs for tail prediction that no previous works have leveraged. Below, we provide a detailed explanation of the innovations introduced in our work, supported by experimental evidence and technical insights.

___Novel Contributions of CAB-KGC___

1. **Head Context Extraction Beyond Descriptions**:
   CAB-KGC utilizes **head context** ($H_c$), a combination of all neighboring *relationships* and *entities* connected to the head entity. The *entities* information helps the model identify how the head entity $h$ is positioned within its local neighborhood (e.g., is it a hub, part of a chain, etc.). The neighbouring *relationships* information offers semantic understanding of the types of relationships the head entity $h$ is generally involved in. This approach captures a richer structural and relational representation of the head entity that NNKGC and similar models fail to consider as these models consider neighboring *entities only*.

2. **Incorporation of Relationship Context**:
   Unlike NNKGC, SimKGC and other existing methods, the CAB-KGC model uniquely incorporates **relationship context** ($R_c$), which includes all entities associated with the operational relationship $r$ in the knowledge graph. This provides the model with a global perspective on the nature of the relation $r$ itself and the patterns or clusters involving the relation $r$. $R_c$ therefore provides __generalization__ ability.  By considering global patterns of the relation, it acts as a regularizer, ensuring that the model aligns with broader relational constraints in the KG. Existing methods, including NNKGC, SimKGC, don't exploit this, thereby failing to fully leverage the structural richness of knowledge graphs.

3. **Independence from Entity Descriptions**:
   Unlike SimKGC and other LLM-based methods, CAB-KGC does not rely on entity descriptions for its operation. This is a critical distinction because many real-world knowledge graphs lack descriptive information for entities. The ability to perform KGC without entity descriptions broadens the applicability of CAB-KGC, making it suitable for datasets where descriptive metadata is sparse or unavailable. For example, we showed that our model demonstrated a substantial improvement in Hits@1 compared to state-of-the-art methods *without using entity descriptions*.

4. **Classification Loss as Training Strategy**:
   The adoption of cross-entropy loss is not merely a substitution for contrastive loss but is better aligned with the KGC task. _Contrastive Loss_ as used in simKGC requires extensive negative sampling, which increases computational cost and introduces noise, especially on large-scale knowledge graphs. _Cross-Entropy Loss_ eliminates the reliance on negative sampling, which is computationally efficient and better suited for large-scale and incomplete knowledge graphs.

5. **Significant Performance Gains**:
   CAB-KGC achieves **state-of-the-art results on multiple datasets**, including FB15k-237, WN18RR, ConceptNet100K  and CoDEx-S with CAB-KGC as top performer in three of four datasets. These results underscore the effectiveness of the proposed model in reliably predicting tail entities across diverse and challenging datasets.

6. **Introduction of EDAS in KGC**:
   Furthermore our work introduces EDAS as a useful evaluation framework for Knowledge Graph Completion (KGC), addressing key limitations of existing metrics. Unlike traditional methods (e.g., MRR, Hits@K), which are often inconsistent, dataset-biased, and assessed in isolation, EDAS provides a unified, aggregated, and dataset-agnostic metric. Ours is the __first application of EDAS in KGC__, demonstrating its ability to holistically balance precision and coverage while penalizing biases inherent to specific datasets. Our empirical results validate its effectiveness, highlighting CAB-KGC’s good performance across multiple datasets—in assigning it the top EDAS rank outcompeting Llama-based DIFT model and others. EDAS thus enhances both the fairness and simplicity of KGC model evaluations.

In summary, the novelty of CAB-KGC lies in its **context-aware design, independence from entity descriptions, and efficient training paradigm**. These contributions directly address limitations in existing methods, making CAB-KGC a significant step forward in KGC research. The experimental results further validate its impact, achieving state-of-the-art performance on challenging benchmarks.


We hope this detailed explanation clarifies the innovative aspects of our work and demonstrates its value to the field.

---

> ### Author Response · Authors · 2024-11-26
> **Justifying EDAS metric for Fair and Robust KGC Evaluation**
>
> Below we hightlight the limitations of currently used metrics, followed by a justification for use of EDAS.
>
> Current KGC models commonly use assessment metrics such as mean rank (MR), mean reciprocal rank (MRR), and Hit@K for evaluating and ranking models, but they come with notable limitations. In particular, most of the KGC models exhibit inconsistencies across these evaluations. For instance, a model may excel in MRR and Hit@3 while performing poorly in Hit@1. These inconsistencies complicate the identification of a __single best-performing approach across different datasets__. MRR, which emphasizes the rank of the first correct prediction, often disproportionately rewards models for achieving high precision at the top of the ranking, ignoring overall coverage. Conversely, Hits@K, which measures the proportion of correct predictions within the top K ranks, can mask issues related to ranking quality by focusing on whether correct answers exist in a limited range, rather than their exact position. Both metrics can be overly influenced by dataset-specific characteristics, such as entity frequency, sparsity, or graph topology, leading to biased or misleading performance assessments. Moreover, these metrics are typically evaluated in isolation, requiring manual interpretation and weighting to understand trade-offs between precision and coverage. Their reliance on single-ground-truth assumptions can further distort results in cases where multiple valid predictions exist but are not explicitly captured in the dataset. These limitations highlight the need for an __aggregated, dataset-agnostic metric__, which balances precision and coverage while reducing biases inherent to individual datasets.
>
> Using the EDAS metric derived from MRR and Hits@K offers below advantages for evaluating and ranking KGC models:
>
> - *Single-Valued Comparative Scoring for Benchmarking:* EDAS provides a single consolidated score to each model across datasets, making it easier to compare models without needing to interpret multiple metrics (e.g., MRR, Hits@1, Hits@3, Hits@10) individually. This is particularly useful for benchmarking across papers, datasets, or use cases. When interpreting MRR and Hit s@K together, there is often ambiguity about how much weight to assign to each metric and how to aggregate metrics from multiple datasets. EDAS inherently balances these contributions to provide a single score for each method.
>
> - *Aggregated and Reliable Assessment Metrics:* EDAS combines the strengths of MRR (focusing on ranking precision) and Hits@K (highlighting prediction coverage) into a unified metric. MRR emphasizes the rank of the first correct prediction, while Hits@K accounts for the presence of correct predictions within a top-K range. EDAS ensures neither is overlooked. By leveraging multiple metrics, EDAS reduces the risk of over-fitting to one specific evaluation criterion, offering a more robust measure of a model's ranking ability.
>
> - *Dataset-Agnostic Evaluations:* EDAS enables the evaluation of model performance across diverse datasets with varying characteristics (e.g., size, density, and graph structure). This uniform metric avoids biases introduced by dataset-specific peculiarities, ensuring that results are comparable. Since MRR and Hits@K can be skewed by dataset construction (e.g., sparsity or entity frequency), aggregating them into a single EDAS score smoothens these effects, providing a holistic measure. This may also help control the false optimism in metrics.  MRR and Hits@K can sometimes produce overly optimistic results in isolation due to dataset idiosyncrasies or trivial solutions (e.g., predicting frequent entities). EDAS penalizes such biases by balancing performance measures across datasets. Furthermore, by aggregating results across multiple datasets, EDAS ensures that a model's strengths are not over-represented due to favourable dataset characteristics, leading to fairer comparisons.
>
>  The first paragraph above on metrics' limitations is added to the paper Introduction while Section 3.1 last paragraph (page 8) of the paper reads the above listed advantages of using EDAS.

---

### Note · Authors · 2024-12-13

**Comment:**

We sincerely thank all reviewers and area chairs for their insightful feedback and thoughtful evaluation of our submission. After careful consideration, we have decided to withdraw our ICLR 2025 submission. We extend our best wishes to every one of you!

**Withdrawal Confirmation:**

I have read and agree with the venue's withdrawal policy on behalf of myself and my co-authors.